# Detailed characterisation of the trypanosome nuclear pore architecture reveals conserved asymmetrical functional hubs that drive mRNA export

Bernardo Papini Gabiatti[1]*, Johanna Krenzer[1], Silke Braune[1], Timothy Krüger[1], Martin Zoltner [2]*, Susanne Kramer [1]*

**1** Biocenter, University of Würzburg, Würzburg, Germany, **2** Department of Parasitology, Faculty of Science, Charles University in Prague, Biocev, Vestec, Prague, Czech Republic

\* bernardo.gabiatti@uni-wuerzburg.de (BPG); martin.zoltner@natur.cuni.cz (MZ); susanne.kramer@uni-wuerzburg.de (SK)

**Data Availability Statement:** All proteomics data have been deposited at the ProteomeXchange Consortium via the PRIDE partner repository

## Abstract

Nuclear export of mRNAs requires loading the mRNP to the transporter Mex67/Mtr2 in the nucleoplasm, controlled access to the pore by the basket-localised TREX-2 complex and mRNA release at the cytoplasmic site by the DEAD-box RNA helicase Dbp5. Asymmetric localisation of nucleoporins (NUPs) and transport components as well as the ATP dependency of Dbp5 ensure unidirectionality of transport. Trypanosomes possess homologues of the mRNA transporter Mex67/Mtr2, but not of TREX-2 or Dbp5. Instead, nuclear export is likely fuelled by the GTP/GDP gradient created by the Ran GTPase. However, it remains unclear, how directionality is achieved since the current model of the trypanosomatid pore is mostly symmetric. We have revisited the architecture of the trypanosome nuclear pore complex using a novel combination of expansion microscopy, proximity labelling and streptavidin imaging. We could confidently assign the NUP76 complex, a known Mex67 interaction platform, to the cytoplasmic site of the pore and the NUP64/NUP98/NUP75 complex to the nuclear site. Having defined markers for both sites of the pore, we set out to map all 75 trypanosome proteins with known nuclear pore localisation to a subregion of the pore using mass spectrometry data from proximity labelling. This approach defined several further proteins with a specific localisation to the nuclear site of the pore, including proteins with predicted structural homology to TREX-2 components. We mapped the components of the Ran-based mRNA export system to the nuclear site (RanBPL), the cytoplasmic site (RanGAP, RanBP1) or both (Ran, MEX67). Lastly, we demonstrate, by deploying an auxin degron system, that NUP76 holds an essential role in mRNA export consistent with a possible functional orthology to NUP82/88. Altogether, the combination of proximity labelling with expansion microscopy revealed an asymmetric architecture of the trypanosome nuclear pore supporting inherent roles for directed transport. Our approach delivered novel nuclear pore associated components inclusive positional information, which can now be interrogated for functional roles to explore trypanosome-specific adaptions of the nuclear basket, export control, and mRNP remodelling.

(Perez-Riverol et al, 2019) with the dataset identifiers PXD055934 (Nup75, 294 Ran, Mex67), PXD047268 (NUP76, NUP96, NUP110), PXD031245 (NUP158, wt control) and PXD059554 (NUP98).

**Funding:** The project was funded by a bilateral GACR/DFG grant (project IDs.: 21-19503J and KR4017/9-1; to M. Z. and S. K., respectively) and the DFG grant KR4017/8-1 to S.K. The funders had no role in study design, data collection and analysis, decision to publish, or preparation of the manuscript.

**Competing interests:** The authors have declared that no competing interests exist.

**Abbreviations:** ALPS, amphipathic lipid-packing sensor; FDR, false discovery rate; IR, inner ring; LAP, lamina-associated protein; LECA, last common eukaryotic ancestor; LFQ, label-free quantification; mRNP, messenger ribonucleoprotein particle; NE, nuclear envelope; NES, nuclear export signal; NLS, nuclear localisation signal; NUP, nucleoporin; OR, outer ring; PCA, principal component analysis; proExM, protein retention expansion microscopy; RBD, RNA binding domain; RMSD, root mean square deviation; RNP, ribonucleoprotein particle; TM, template modelling; UExM, ultrastructural expansion microscopy.

## Introduction

Nuclear pores penetrate the double-membrane of the nucleus and serve as an essential gateway for the exchange of proteins, RNAs and ribosomes between the nucleoplasm and the cytoplasm. They are among the largest macromolecular complexes in nature with more than 500 copies of approximately 30 different nucleoporins (NUPs) that form 8 identical protomers (spokes) [1–4]. Each spoke is connected to the nuclear envelope (NE) as well as to the neighbouring spokes, resulting in multiple concentric rings: the inner ring (IR) at the centre of the pore is flanked by 2 outer rings (ORs) at the cytoplasmic site (cytoplasmic ring) and nuclear site (nuclear ring). The outer rings are composed of large Y-shaped protein complexes, called the Nup84 complex in yeast and the NUP107 complex in humans [4]. This pore core structure is extended by the nuclear basket at the nuclear site and the cytoplasmic filaments at the cytoplasmic site. NUPs can be divided into 3 classes: (i) structured NUPs that form the scaffold of the pore, with structural features being limited to beta propellers, coiled coil and alpha-helical solenoids; (ii) pore membrane proteins (POMs) that anchor the pore in the nuclear envelope via transmembrane regions; and (iii) non-structured, intrinsically disordered NUPs, that contain FG (phenylalanine and glycine) repeat motifs and provide a diffusion barrier at the central channel of the pore by phase separation [5]. While smaller molecules can pass by diffusion, the transport of larger molecules, such as most RNAs, ribonucleoprotein particles (RNPs), pre-ribosomes and most proteins, requires energy and depends on transporters. Protein transport, as well as the transport of micro-RNAs and tRNAs is mediated by importins and exportins of the karyopherin family [6]. These transporters recognise and bind nuclear localisation signals (NLSs) or nuclear export signals (NESs) of their cargo and shuttle it within the phase-separated central channel of the pore by interacting with the FG-repeat NUPs [6]. This transport is energised by the RanGTP/RanGDP gradient across the nuclear envelope maintained by the chromatin-bound guanine nucleotide exchange factor RCC1 and the cytoplasmic-localised proteins Ran-GTPase-activating protein RanGAP and RanBP1 [6]. Importins bind cargo and Ran-GTP mutually exclusively, while exportins bind Ran-GTP and cargo cooperatively, thus allowing selective release of cargo either in the nucleus or cytoplasm, respectively, driven by GTP hydrolysis cycles of Ran [6]. The vast majority of messenger ribonucleoprotein particles (mRNPs) are not exported by karyopherins but use the heterodimeric Mex67/Mtr2 complex (NXF1 or TAP/NXT1 in humans) instead [7–9]. The energy is provided by at least 2 RNA helicases, Sub2 and Dbp5 in yeast and UAP56 and DDX19 in human, that assemble and disassemble the Mex67/Mtr2/mRNA export complex in the nucleus and in the cytoplasm, respectively, in events known as nuclear and cytoplasmic mRNP remodelling [10]. In ophistokonts, 2 asymmetric pore components, the basket and the cytoplasmic filaments, ensure directionality of RNP transport. The basket of *Saccharomyces cerevisiae* consists of Nup1, Nup2, Nup60, Mlp1/2, and Pml39 and in metazoan of NUP153 (orthologue to yeast Nup1/60), NUP50 (orthologue to yeast Nup2), TPR (orthologue to yeast Mlp1/2), and ZC3H1 (orthologue to yeast Pml39). Nup60 anchors the yeast nuclear basket to the Y-complex of the nuclear outer ring [11]. Nup1 and NUP153 anchor the TREX-2 (3 prime repair exoribonuclease 2) complex to the pore in yeast [12] and humans [13], respectively. TREX-2 in yeast/human consists of the large scaffolding protein Sac3/GANP bound to Thp1/PCID2, Sem1/DSS1, and Sus1/ENY2, and, in yeast only, to Cdc31. This complex functions in recruiting the mRNP to the pore by direct interaction of the Sac3 N-terminal region with the mRNA-loaded Mex67 [14]. The cytoplasmic filaments are heterotrimeric complexes of Nup82/NUP88, Nup159/NUP214, and Nsp1/NUP62 (yeast/human); Nup159/NUP214 recruit the mRNA remodelling helicase Dbp5/Ddx19 with its cofactor Gle1 [15,16].

Structure and composition of nuclear pore complexes has been characterised in a range of organisms, including *S. cerevisiae* [17,18], humans [19], the thermophilic fungus *Chaetomium thermophilum* [20], and the algae *Chlamydomonas reinhardtii* [21]. While the general structure of the pore, in particular the inner ring structure, is highly conserved, the more peripheral structures can differ between organisms and even within the same organism [22]. Yeast for example has up to 3 pore variants that differ in the number of nuclear outer rings and in the presence or absence of a basket [23–26]. Trypanosomes have separated from the eukaryotic main branches very early and their nuclear pore architecture is thus an important stepping-stone towards a better understanding of pore evolution. In particular, structural differences would help to unravel which pore features constitute organism-specific adaptations and which have been present in the LECA (last common eukaryotic ancestor) [27,28].

In *Trypanosoma brucei*, 22 NUPs were initially identified based mostly on predicted structural similarities to human and yeast NUPs, as the sequences are poorly conserved, and pore localisation was confirmed by GFP-tagging [29]. This served as foundation for a hallmark follow-up study that has defined the sub-complexes, quaternary structure, and pore-associated proteins by a large set of immunoprecipitations with multiple baits from cryomilled samples, combined with immunogold electron localisation and in silico prediction tools [30]. Similar to all other eukaryotes studied so far, the inner ring is mostly conserved [22,30,31], with the one exception of the membrane anchoring mechanism: *T. brucei* lacks orthologues to all POMs of opisthokonts. Instead, TbNUP65 has evolved a C-terminal transmembrane helix to connect to the nuclear envelope [30], replacing the amphipathic lipid-packing sensor (ALPS) motif used by its opisthokont orthologues ScNup53/59 and HsNUP35. Two additional POM candidates, with transmembrane helixes, were recently identified within the interactome of lamin-like proteins [32]. The structured outer ring complex (Y-complex) was clearly defined in multiple affinity purifications to consist of TbNUP158, TbSEC13, TbNUP41, TbNUP82, TbNUP89, TbNUP132, TbNUP152, and, likely, TbNUP109 [30]. This complex, named NUP89 complex, is the equivalent to the yeast outer ring complex Nup84 (NUP107 in human) and is mostly conserved, with some lineage-specific variations in the β-propeller proteins [30]. Three FG-NUPs, NUP64, NUP75, and NUP98, are unique to trypanosomes and part of one complex with unknown localisation [30]. There were 2 major unexpected outcomes from this study: (i) no asymmetrically localised NUPs were identified, with the exception of the basket proteins NUP110 and NUP92, suggested as putative homologues to yeast Mlp1 and Mlp2. Even TbNUP76, which was co-isolated with TbMEX67 and has structural homology to the cytoplasmic site-specific yeast Nup82 that has a function in mRNA export [33,34] was predicted at both outer rings by immunogold labelling; (ii) the authors could not identify any homologue to the cytoplasmic mRNA remodelling enzyme, the DEAD-box RNA helicase Dbp5. Instead, they found co-purification in high-stringency conditions between the conserved mRNA transporter TbMEX67 with TbRan, TbRanBP1, and a putative *T. brucei* RanGAP, indicating that mRNA export may be fuelled by the Ran system.

Meanwhile, many additional nuclear pore-localised proteins were identified, primarily by the genome-wide localisation database TrypTag [35], of which most remain functionally uncharacterised.

We were puzzled by the absence of asymmetric NUPs at the outer rings, which are viewed as key determinants underpinning directed transport of macromolecules. We therefore revisited the ultrastructure of the trypanosome nuclear pore using a novel, powerful combination of expansion microscopy and proximity labelling techniques. Our approach indeed identified a set of asymmetric components and we employed these as markers to map all 75 nuclear pore-localised proteins reported by TrypTag [35]. Altogether, we provide an updated, comprehensive map of the pore and its associated proteins, including proteins of the Ran GTPase

transport system. We describe many novel proteins at the nuclear site of the pore, most of these trypanosome-unique, including 3 potential TREX-2 complex proteins. We find the NUP76 complex proteins, NUP76, NUP140, and NUP149, exclusively at the cytoplasmic site and demonstrate a conserved function of NUP76 in mRNA export, while NUP140 and NUP149 are unique to trypanosomes, and lack any conserved binding site for Dbp5, consistent with the absence of this RNA helicase. Our data, combined with the data of [30], support a model of the trypanosome pore with a conserved core structure, but with a fundamentally different mRNA remodelling platform at the cytoplasmic site and many trypanosome-unique proteins at the basket site that await functional characterisation.

## Material and methods

### Bioinformatics

All sequences were retrieved from TriTrypDB between 2021 and 2024 [36]. InterPro was used for domain search based on sequence [37]. Homology search based on primary and predicted secondary alignments was done with Phyre2 [38]. Tertiary alignments of Trypanosomatid-optimised predicted AlphaFold2 models [39] were carried out with Foldseek [40]. Foldseek searches were performed on the web server (https://search.foldseek.com) covering all available databases (AlphaFold/Proteome v4, AlphaFold/Swiss-Prot v4, AlphaFold/Swiss-Prot v4, BMFD 20240623, CATH50 4.3.0, Mgnify-ESM30 v1, PDB100 20240101, and GMGCL 2204) with Mode 3Di/A. The outputs from Foldseek including the superimposed structures, the values of sequence identity, RMSD (root mean square deviation), TM (template modelling score), qTM and tTM (TM scores normalised by query and template length, respectively) values were retrieved. All structures were predicted using AlphaFold2-Multimer-v2.3.1 [41,42] through the ColabFold version 1.5.3 with Mmseq2 (UniRef+Environmental) with 5 recycles and 5 models (doi:10.1038/s41592-022-01488-1). Predicted structures were visualised with ChimeraX [43]. Heatmaps were generated with the ComplexHeatmap package [44] in R. The $t$ test difference values from the affinity purifications (detailed below) were fed in and clustered with a Pearson distance method (option cluster_rows = TRUE, cluster_columns = FALSE). The $t$ test difference values are represented as a colour scale and the colouring was made by the package. pLDDT plots of local prediction confidence over the protein length shown near the heatmaps are available and were retrieved from the Trypanosomatid-optimised AlphaFold2 database [39].

### *Trypanosoma* cells

*Trypanosoma brucei* Lister 427 procyclic cells in logarithmic growth were used for all experiments. Cells were grown in SDM-79 supplemented with 5% (v/v) FCS and 75 μg/ml hemin at 27˚C, 5% $CO_2$, and appropriate drugs [45]. Drugs used for transgenic cells were G418 disulfate (15 μg/ml), blasticidine S (10 μg/ml), puromycin dihydrochloride (1 μg/ml), hygromycin B (25 μg/ml), and phleomycin (2.5 μg/ml); these concentrations were used for maintenance and doubled during the actual selection process after transfection. Growth was measured by sub-culturing cells daily to $10^6$ cells/ml and measuring densities 24 h later using a Coulter Counter Z2 particle counter (Beckman Coulter) over 5 days.

Transgenic trypanosomes were generated by standard procedures. Endogenous tagging with TurboID-Ty1, TurboID-HA, 3xHA, 4xTy1, and *Os*AID-3xHA was done using a PCR-based method and the pPOTv7 system [46]. TurboID-Ty1, 3xHA, and 4xTy1 customisations of the pPOTv7 were made in this work; 25 μl of PCR reaction (PrimeSTAR MAX (Takara)) was used for transfections. The PCR product was precipitated with isopropanol, washed once with 70% ethanol in a sterile hood, resuspended in 10 μl of sterile ddH2O, and mixed with $10^7$

cells in 400 μl of transfection buffer [47]. Transfections were performed with Amaxa Nucleofactor IIb (Lonza Cologne AG, Germany, program X-001) using BTX electroporation cuvettes (45–0125). Cells were recovered in 25 ml SDM-79 supplemented with 20% FCS for 18 h and diluted 1:4 in 75 ml SDM-79 supplemented with 20% FCS. Relevant drugs were added and cells plated in four 24-well plates (1 ml/well). Drug-resistant populations were analysed after 10 days and confirmed with western blotting or diagnostic PCR from genomic DNA [48]. For ectopic, inducible expression of MEX67 fused to TurboID-Ty1 at the C-terminus, its open reading frame was cloned in frame with TurboID-Ty1 in a genetic cassette containing an EP procyclin promoter controlled by 2× Tet operator flanked by sequences for integration to the rRNA locus [49].

For auxin-inducible degron experiments, 50 μm 5-Ph-IAA (MedChem Express, HY-134653) was added to the cultured cells from a 50 mM stock in DMSO. The auxin system was kindly provided by the laboratory of Mark Carrington (University of Cambridge, United Kingdom) and is described in [50].

## Plasmids and PCR products

All plasmids and PCR products used in this study are listed in S1 Table. pPOTv7 variants for TurboID-Ty1, 3xHA, 4xTy1 tagging were generated by sub-cloning the respective tag sequence in the BamHI/HindIII sites. pPOTv7 *Os*AID-3xHA was generated in [50]. Note that the TurboID-Ty1 and TurboID-HA tags will be referred only as TurboID in text and figures to avoid confusion; the Ty1 and HA tags were solely used to control cell lines by western blot, not for imaging.

## Western blot and antibodies

Western blots were done using standard methods. Primary antibodies used for detection of proteins were rat anti-HA (3F10, Roche) (1:1,000), anti-Ty1/BB2 ([51] hybridoma supernatant 1:1,000) and anti-*T. brucei* PFRA/B (L13D6) (1:10,000) [52]. Secondary antibodies were IRDye 680 RD and 800 CW rat and mouse anti-goat (LI-COR) (1:30,000). Biotinylated proteins were detected with Streptavidin-IRDye 680 LT (LI-COR) (1:10,000). Blots were scanned with the Odyssey Infrared Imaging System (LI-COR Biosciences, Lincoln, Nebraska, United States of America).

## Protein retention expansion microscopy (proExM) and ultrastructural expansion microscopy (UExM)

The proExM and UExM methods were performed as previously described in [53], with the following minor modifications for UExM: the primary and secondary antibody labelling reactions were done in 6-well plates with 1 ml of antibody diluted in PBS-T (PBS with 0.1% Tween20 (v/v)) containing 3% BSA (w/v). Primary antibodies were incubated overnight at 37˚C with agitation and secondary antibodies for 3 h at 37˚C with agitation. The plates were slightly tilted to ensure that the whole gel piece was covered.

## Microscopy and quantification of microscopy data

For all fluorescence microscopy experiments, images were acquired using a fully automated iMIC microscope (TILL Photonics) equipped with a 100×, 1.4 numerical aperture objective (Olympus, Japan) and a sensicam qe CCD camera (PCO, Germany). Z-stacks (75, 100, or 150 slices, 140 nm step size) were recorded. Exposure times ranged between 50 and 100 ms for DAPI and 400 to 800 ms for all other fluorophores. Image stacks were deconvolved with the

Huygens Essential software v24.04 (SVI, Hilversum, the Netherlands). Deconvolution parameters were kept constant for all images, except for the number of iterations which were optimised depending on the signal intensity and background. To correct aberrations due to the refractive index mismatches occurring at different depths of the gel specimens, the varPSF function of Huygens Essential was used, which calculates the PSFs according to depth position. After deconvolution, images were corrected for the chromatic shift aberration between the green and red channel in 3 dimensions. The Huygens Chromatic Aberration Corrector was used with image stacks of TetraSpeck fluorescent microspheres (T7279, Thermo Fisher Scientific) as template. Fiji [54] was used for figure generation.

## Streptavidin affinity purification and LC MS/MS analysis

Affinity purification of biotinylated proteins followed by tryptic digest and peptide preparation were done as described [55], except that 1 mM biotin was added to the on-beads tryptic digests, to improve the elution. Eluted peptides were analysed by liquid chromatography coupled to tandem mass spectrometry (LC-MS/MS) on an Ultimate3000 nano rapid separation LC system (Dionex) coupled to an Orbitrap Fusion mass spectrometer (Thermo Fisher Scientific).

Spectra were processed using the intensity-based label-free quantification (LFQ) in Max-Quant version 2.1.3.0 [56,57] searching the *T. brucei brucei* 927 annotated protein database (release 64) from TriTrypDB [58]. Analysis was done using Perseus [59] essentially as described in [60]. Briefly, known contaminants, reverse hits (decoy sequences for calculating the false discovery rate (FDR)) and hits only identified by a modification site were filtered out. LFQ intensities were $\log_2$-transformed and missing values imputed from a normal distribution of intensities around the detection limit of the mass spectrometer. A Student's *t* test was used to compare the LFQ intensity values between the duplicate samples of the bait with untagged control (WT parental cells) triplicate samples. The $-\log_{10}$ *p*-values were plotted versus the *t* test difference to generate multiple volcano plots (Hawaii plots). Potential interactors were classified according to their position in the Hawaii plot, applying cut-off curves for significant class A (SigA; FDR = 0.01, s0 = 0.1) and significant class B (SigB; FDR = 0.05, s0 = 0.1). The cut-off is based on the FDR and the artificial factor s0, which controls the relative importance of the *t* test *p*-value and difference between means (at s0 = 0 only the *p*-value matters, while at non-zero s0 the difference of means contributes). Perseus was also used for principal component analysis (PCA), the profile plots and to determine proteins with similar distribution in the plot profile using Pearson's correlation. All proteomics data have been deposited at the ProteomeXchange Consortium via the PRIDE partner repository [61] with the data set identifiers PXD055934 (Nup75, Ran, MEX67), PXD047268 (NUP76, NUP96, NUP110), PXD031245 (NUP158, wt control), and PXD059554 (NUP98).

## Results

### Expansion microscopy identifies novel asymmetric components of the trypanosome pore

We revisited the trypanosomatid nuclear pore architecture with expansion microscopy. Therefore, we expressed target proteins fused to a small peptide epitope-tag (3xHA or 4xTy1) to allow immunofluorescence detection via anti-HA or anti-Ty1. In some experiments, we expressed the target protein fused to the biotin ligase TurboID [62], followed by the detection of the biotinylation of the bait and proximal proteins with fluorescent streptavidin (= streptavidin imaging). We had previously shown that labelling with streptavidin increases the signal

with no obvious loss in resolution, which is essential since expansion microscopy causes a massive reduction in antigen density [53]. Even more importantly, streptavidin readily labels proteins within phase-separated areas, such as the nuclear pore central channel, that we found largely inaccessible to antibodies [53]. Since TurboID will not only auto-biotinylate the bait but also adjacent proteins, there is the possibility that the streptavidin signal may not reflect the true localisation of the bait. Hence, throughout this work, we have confirmed all major findings derived from streptavidin labelling with orthogonal methods, such as immunofluorescence, mass spectrometry, and/or vice-versa labelling. All fusion proteins were expressed from the endogenous loci to avoid major changes in gene expression.

First, we used protein retention expansion microscopy (proExM), a method that expands cells after protein labelling [63]. We experimentally determined the expansion factor as 3.6 and confirmed the isotropic expansion of the nucleus (S1 Fig). We initially concentrated on the proteins of the NUP76 complex (NUP76, NUP140, and NUP149) as these were previously shown to co-precipitate with the trypanosome mRNA export factor MEX67 [30]. These NUPs were fused to either N- or C-terminal peptide tags (NUP76::3xHA, NUP140::4xTy1, 3xHA:: NUP149). In the same cell lines, we co-expressed the nuclear basket protein NUP110 with an N-terminal fusion to a different epitope tag (4xTy1 or 3xHA). Upon dual labelling with anti-Ty1 and anti-HA, we carried out expansion and imaging. All 4 proteins resolved as single dots located at the nuclear periphery. The signals from the NUP76-complex proteins were in all cases clearly separated from the NUP110 signal towards the cytoplasmic site of the pore (Figs 1A and S2). Notably, we observed for every dot signal originating from the NUP76 complex a corresponding NUP110 dot, indicating that trypanosomes, unlike yeast [26], do not have basket-less pores. The median, expansion-factor corrected distance to NUP110 exceeded for all 3 proteins 120 nm (129±17 nm for NUP76, 120±22 nm for NUP140, and 137±18 nm for NUP149 with $n > 100$). The large distance of the NUP76 complex proteins to the basket protein NUP110 and the absence of "double-dots" for the NUP76-complex strongly indicate asymmetric localization of the NUP76 complex exclusively at the cytoplasmic site of the pore. The data disagree with previous observations derived from immuno-electron microscopy that place NUP76, symmetrically, to both outer rings [30].

We were concerned that the observed sole cytoplasmic localisation of the NUP76 complex proteins is a technical artifact caused by (i) an insufficient resolution of the proExM method; (ii) a non-isotropic expansion across the nuclear membrane; or (iii) reduced accessibilities of antibodies to the nuclear ring in comparison to the cytoplasmic ring. To investigate, we applied ultrastructural expansion microscopy (UExM) [64], which offers a higher resolution because the antibody labelling is applied after the expansion and the linkage error (distance between the fluorophore of the secondary antibody and the target protein) is therefore not expanded. UExM has been successfully used in trypanosomes [53,65–67], and we achieved an expansion factor of 4.2-fold with isotropic expansion of the nucleus (S1 Fig). To prove that UExM provides the resolution to resolve the nuclear outer ring, inner ring, and cytoplasmic outer ring, we first imaged NUPs that are conserved across eukaryotes. As an outer ring marker, we chose NUP89, the trypanosome orthologue to the outer ring Y-complex component of yeast Nup84/85 (NUP107/75 in human) [30]. As an inner ring marker, we selected NUP96, the conserved trypanosome orthologue to *S. cerevisiae* Nic96. We co-expressed NUP89::TurboID with NUP96::3xHA. UExM with fluorescent streptavidin and anti-HA readily resolved NUP89 as double dots at the nuclear periphery, sandwiching the single dot signal of NUP96 (Fig 1B), proving that the resolution of the method is sufficient to distinguish these different subregions of the pore. However, as the NUP96 signal was weak, we searched for a better inner ring marker. We tested NUP64, expressed as C-terminal 3xHA fusion, a trypanosome-unique FG-repeat NUP previously identified as a multi-complex NUP localised mostly

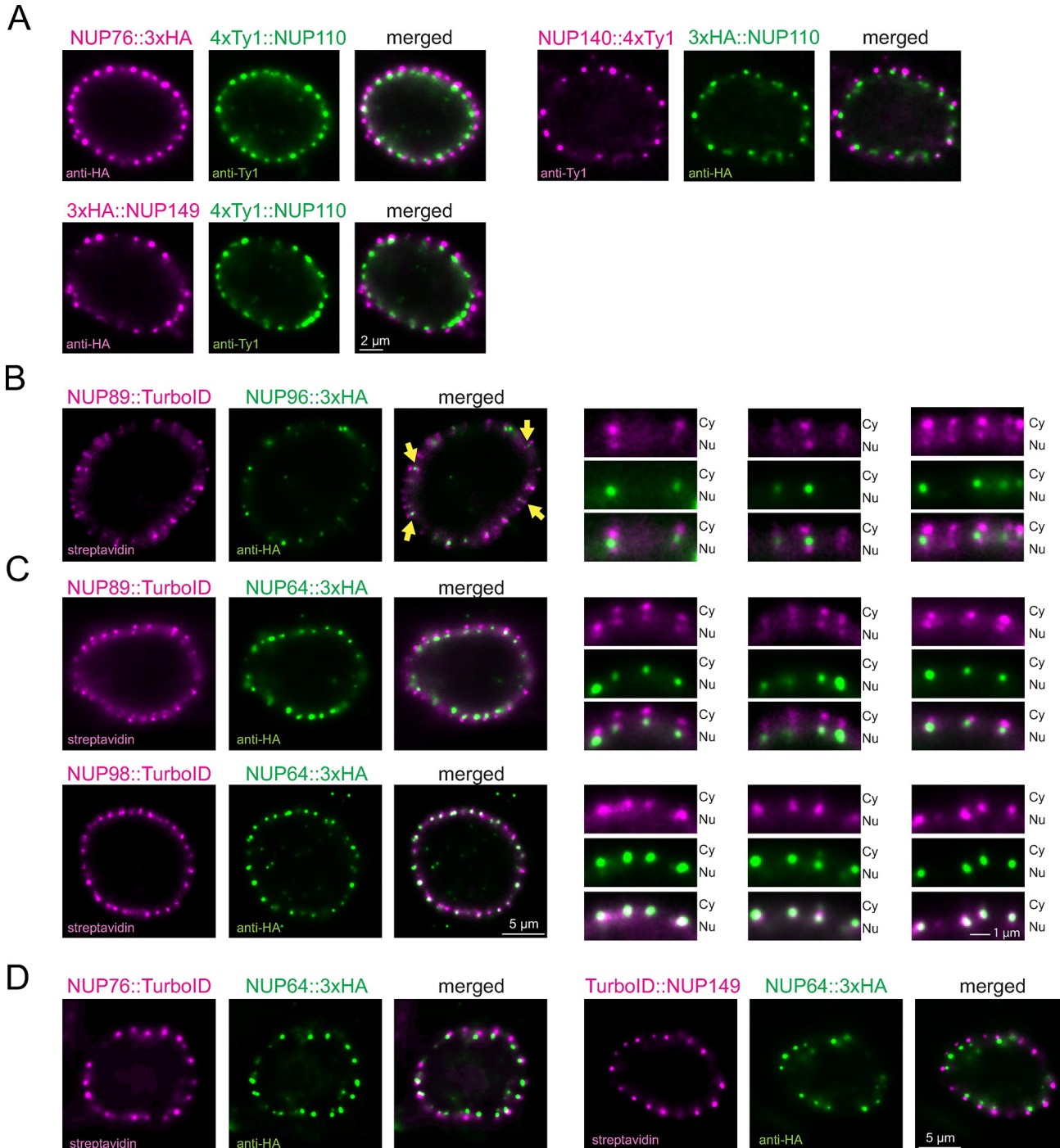

**Fig 1. Expansion microscopy identifies novel asymmetric pore proteins.** **(A)** proExM of cell lines co-expressing epitope-tagged versions of NUP76, NUP140, or NUP149 in combination with NUP110. Images were deconvolved (20 iterations for NUP149 and NUP76; 60 iterations for NUP140) and single planes of the nuclei are shown. Further images are shown in S2 Fig. **(B, C)** UExM of cells expressing proteins fused to TurboID or 3xHA, as indicated. Labelling was done with fluorescent streptavidin and with anti-HA. Images were deconvolved (20, 20, 40 iterations for NUP89/NUP96, NUP89/NUP64, and NUP98/NUP64, respectively). A single plane image of an entire nucleus is shown on the left and 3 enlarged regions from the same or another nucleus are shown on the right (Cy = cytoplasm, Nu = nucleus). For NUP89/NUP96, yellow arrows point to pores that are in a suitable focal plane to see the 2 NUP89 dots sandwiching the NUP96 dot. **(D)** UExM of lines co-expressing TurboID-tagged versions of NUP76 or NUP149 with NUP64-3xHA. Labelling was done with fluorescent streptavidin and anti-HA. Images were deconvolved with 60 and 20 iterations for NUP76/NUP64 and NUP149/NUP64, respectively. A single plane image of one nucleus is shown. proExM, protein retention expansion microscopy; UExM, ultrastructural expansion microscopy.

to the centre of the pore [30]. To our surprise, the resulting single NUP64 dot signal was not sandwiched by the 2 outer ring dots of NUP89 but instead co-localised solely with the NUP89 dot at the nuclear site of the pore (Fig 1C). Likewise, the streptavidin signal of a C-terminal TurboID fusion of TbNUP98, known to form a complex with NUP64 [30], resolved as single dots that colocalised exclusively with the NUP64::3xHA dots at the nuclear site (Fig 1C). The data indicate an asymmetric, exclusive nuclear site localisation of NUP64 and NUP98 (Fig 1C).

Having confirmed that UExM has the resolution to distinguish proteins located at the cytoplasmic site outer ring from proteins located at the nuclear site outer ring, we reassessed the localisation of the NUP76 complex. We confirmed the sole cytoplasmic localisation of the NUP76 complex by co-staining fusions of this complex to either 3xHA (S3 Fig) or TurboID (Fig 1D) with C-terminal 4xTy1 or 3xHA fusions of our newly identified nuclear site marker NUP64.

In summary, we discovered 5 asymmetric proteins of the trypanosome nuclear pore complex, previously assumed to be symmetrically distributed: NUP76/NUP140/NUP149 at the cytoplasmic site and NUP64/NUP98 at the nuclear site. With the exception of NUP76, which is the structural orthologue to yeast NUP82 and human NUP88 [30], all novel asymmetric NUPs are trypanosome specific.

## A proximity map of the trypanosome nuclear pore

The novel availability of asymmetric NUPs prompted us to use mass spectrometry data from TurboID proximity labelling experiments, to generate a proximity map of the entire trypanosome nuclear pore. We used 7 available LC-MS/MS data sets from previous streptavidin-affinity purifications, namely N- and C- terminal TurboID fusions of NUP110 (basket), NUP96 (inner ring), and NUP76 (cytoplasmic outer ring) [53] and NUP158 (outer ring) with C- terminal TurboID fusion [55]. In addition, we generated a new LC-MS/MS data sets for NUP98 that we have identified to be at the nuclear-site of the pore by expansion microscopy, and also for NUP75, which was previously identified to be in a complex with NUP98 and NUP64 [30].

Of all the proteins that were labelled by these baits, we initially concentrated on proteins that were previously identified as NUPs, based on predicted structural similarities [29] and/or affinity purification [30]. For each data set, we colour-coded the enrichment of the NUP proteins based on *t* test difference to a wild-type control. Then, the NUPs were sorted by hierarchical clustering applying a Pearson distance method (Fig 2A). For each NUP, we included the pLDDT plots [39] to indicate confidence of the predicted structure, which in most cases correlates to structured (high confidence) and unstructured (low confidence, mostly FG-repeats) regions. The majority of the NUPs separated into 3 clearly distinct main clusters. The first cluster contained the lamina protein NUP2 [68] and NUP64/NUP98 that localised exclusively to the nuclear site of the pore by expansion microscopy (Fig 1C). The second cluster largely consisted of NUPs previously identified as inner ring NUPs [30] while the third was dominated by NUPs previously assigned to the outer ring [30]. Five NUPs were manually added to the clusters using positional information from [30], because of poor labelling by only 2 baits or less (NUP62, NUP119, NUP110, NUP92) or labelling by all baits (NUP132). For 6 proteins, the labelling was too weak (NUP152, SEC13A, SEC13B, NUP41, NUP48) or too diverse (GLE2) to confidently assign the proteins to a certain region of the pore.

For the vast majority of NUPs, the proximity map confirmed the previous assignments of the NUPs from affinity capture experiments [30]. Only 2 NUPs exhibited ambiguous placement in the proximity map (bold in Fig 2A). The outer ring NUP132 is labelled strongly by all 8 baits, including strong labelling by the basket proteins NUP110 and NUP92, suggesting

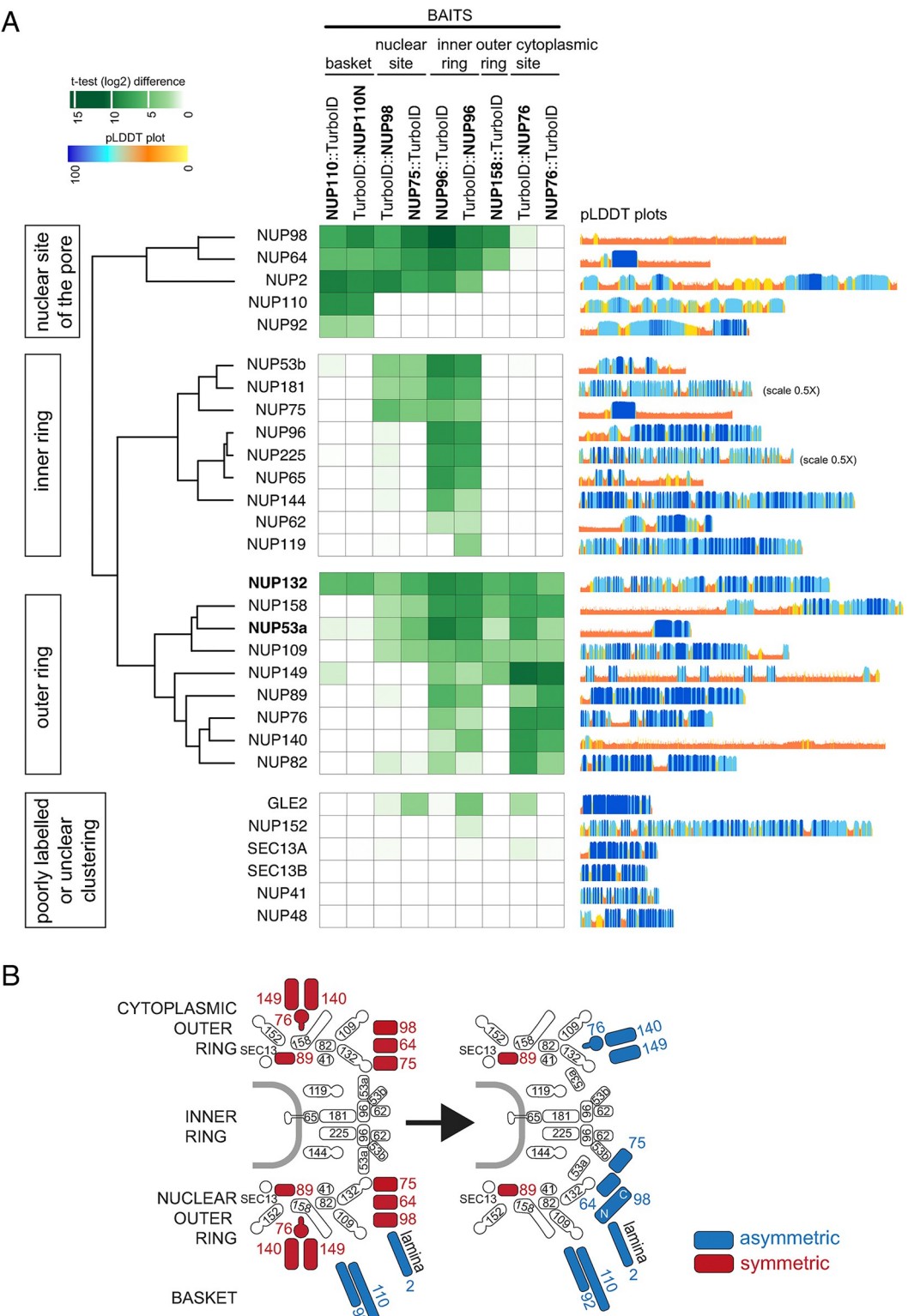

**Fig 2. A proximity map of the trypanosomatid nuclear pore. (A)** Mass spectrometry data (*t* test difference values) from proximity labelling experiments were used to create a heat map of the trypanosome nuclear pore. A range of N- or C-terminally tagged TurboID fusions served as baits and the labelling (proximity) of most nuclear pore proteins is shown as a tree. Some NUPs were manually added to the tree using data of [30]. pLDDT plots from Trypanosomatid-optimised AlphaFold2 models [39] are shown on the right. Details on the mass spectrometry data can be found in S1 Table. **(B)** The

model of the trypanosome nuclear pore changes, with the discovery of 5 novel asymmetrically localised proteins. NUP, nucleoporin.

extensions of NUP132 towards the basket region. Further, the presumed inner ring NUP53a is also labelled by all baits: the labelling by the inner ring bait NUP96 is the strongest, but there is also strong labelling by the cytoplasmic-specific NUP76 and also by nuclear site-specific NUPs, indicating that NUP53a may be at the inner ring but reaching out to the outer rings.

NUP98 and NUP64 unequivocally grouped with the basket/nuclear site (basket and inner ring) and were labelled by NUP110 and NUP96 baits, but not by the cytoplasmic NUP76, in line with our proExM and UExM data (compare Fig 1B and 1C). To our surprise, NUP75, which shares 46% sequence identity with NUP64 and associates with both NUP98 and NUP64 [30], was placed to the inner ring and was not labelled by NUP110. Moreover, the outer ring protein NUP158 strongly labelled NUP64 and slightly less NUP98, but not NUP75, further supporting the absence of NUP75 from the outer rings [53]. When NUP75 and NUP98 were used as baits, both showed the strongest labelling with each other and with NUP64 (Fig 2A), consistent with these 3 proteins forming a complex, as previously suggested [30]. Moreover, as expected, neither protein labelled the NUP76 complex, which, incidentally, is orthogonal evidence for the NUP76 complex being cytoplasmic. Interestingly, neither protein labelled NUP110, which was expected for NUP75, but not for NUP98, which is itself labelled by NUP110. Perhaps, the C-terminus of NUP98 is distant to NUP110, while the N-terminus is close. Our data suggest a model of an asymmetric NUP98/64/75 complex reaching from the nuclear outer ring to the inner ring, with NUP98 and NUP64 being located at the outer nuclear ring and NUP75 at the inner ring. Data from previous affinity isolation experiments with NUP98, NUP64, and NUP75 show marked differences between the interactomes of NUP98/64 and NUP75, including the exclusive absence of NUP110 from the NUP75 interactome, in full agreement with our data [30].

The outer-ring cluster is divided into 2 subclusters. Proteins of both clusters are labelled by the inner ring bait NUP96 and by the cytoplasmic-site-specific NUP76. Proteins of the first cluster (NUP158, NUP53a, and NUP109) are additionally labelled by the nuclear site-specific proteins NUP98 and NUP75 and by NUP158, while proteins of the second cluster are not (with the one exception of NUP149, which is labelled by NUP158). We believe that this clustering reflects differences in proximity between these 2 protein groups within the outer rings. We can exclude the interpretation that the clustering of NUP89 and NUP82 with the cytoplasmic site-specific NUP75/NUP140/NUP149 proteins means, that NUP89 and NUP82 are cytoplasmic-site specific too: NUP89 is present in both outer rings (Fig 1B and 1C) and both NUP89 and NUP82 are weakly labelled by the nucleoplasmic-specific NUP75 and NUP98.

In summary, the proximity map accurately predicts the localisations for the vast majority of NUPs. Importantly, it offers orthogonal (tag-independent) validation of the asymmetric localisation of NUP98 and NUP64 to the nuclear site of the pore and of the NUP76 complex proteins to the cytoplasmic site of the pore, confirming the data of the expansion microscopy. A new model of the pore, highlighting the asymmetric components, is shown in Fig 2B. Thus, our proximity map has the potential to predict localisations of proteins with sub-pore size resolution, which prompted us to look at all 44 proteins that have nuclear pore localisation according to TrypTag [35] but are not annotated as NUPs.

## Mapping the Ran-based mRNA export system to the pore

First, we concentrated on all nuclear pore-localised proteins that are involved in mRNA export: MEX67 [69], Mtr2, MEX67b [70] and, as postulated [30,71], Ran, RanGAP, the

putative RanGDP importer NTF2 and 2 Ran-binding proteins, RanBP1 and RanBPL [72]. The proximity map places RanGAP and RanBP1 to the cytoplasmic site of the pore, while RanBPL localisation is predicted at the nucleoplasmic site (Fig 3A). For the transporters MEX67, MEX67b and Ran, the labelling was less confined to a specific site. For Mtr2 and NTF2, we obtained no labelling, likely due to their small size which is problematic in BioID [55]. As a control, we included data of vice versa TurboID experiments with MEX67 and Ran as baits (Figs 3A and S4 and S2 Table): both proteins label proteins at both sides of the pore, consistent with shuttling.

Next, we determined the localisation of MEX67, Ran, RanGAP, RanBP1, and RanBPL by UExM, expressing TurboID fusions in a cell line that expressed NUP64::3xHA as a nucleoplasmic-site marker (Fig 3B and 3C). RanGAP and RanBP1 resolved as single dots, well distanced from the NUP64 dots towards the cytoplasmic site, while the RanBPL signals overlapped with the NUP64 signals at the nuclear site, in full agreement with the proximity map. The biotinylation signal of the 2 proteins with suspected shuttling activity, Ran and MEX67, resolved as large cytoplasmic dots and smaller nuclear dots, connected by a string-like signal reaching through the pore. For MEX67, we observed that these bone-shaped signals were more pronounced when the gene was 8-fold overexpressed from an ectopic locus (Fig 3B, images on the right) which only slightly impaired growth (S5 Fig). For Ran, MEX67, and RanBPL, we observed an additional signal at the nucleolus, which is defined by the reduction in DAPI stain (S6A Fig), and a minor signal in the nucleoplasm. For RanBPL, the nucleolar signal was stronger than the signal at the pores, while for Ran and MEX67 (at endogenous expression levels) the nuclear pore signal was dominant. We attempted to confirm the nucleolar localisation by direct immunofluorescence instead of streptavidin imaging. The nucleolus is challenging to label with antibodies [53], but for MEX67::4xTy1 we could get a weak, but distinct nucleolar antibody signal (S6B Fig). The functional implications of the nucleolar localisation of Ran, MEX67, and RanBPL are not fully understood in trypanosomes but not unexpected, as in ophistokonts Ran and Mex67 participate in pre-ribosome transport [73].

For the shuttling proteins Ran and MEX67, we confirmed the streptavidin-based imaging data by the LC-MS/MS data upon streptavidin enrichment (Figs 3D and S4 and S2 Table). Both MEX67 and Ran strongly labelled FG-NUPs lining the inner pore channel, consistently reflecting their movement across the pore. The NUPs with the strongest labelling were the asymmetric NUPs on both sides of the pore: NUP149 at the cytoplasmic site and NUP98, NUP64 and the lamin-like protein NUP2 at the nuclear site (red arrows in Fig 3D). This strongly suggests that both proteins, Mex67 and Ran, would have binding sites at both sides of the trypanosome pore, analogous to human Ran [74–77]. There was weak labelling of structured NUPs, in agreement with the rather poor labelling of MEX67 and Ran by NUP76, NUP96 and NUP110 and NUP158 in the heat map (Fig 3A). Preferential labelling of asymmetric FG NUPs over structured NUPs has also been shown for human karyopherins tagged with the biotin ligase BirA* [78].

In conclusion, our proximity map predicted the localisation of all non-shuttling Ran system components confidently and in agreement with streptavidin imaging in UExM. RanGAP is at the (expected) cytoplasmic site, together with RanBP1. RanBPL had not been previously mapped, but is unequivocally placed to the nuclear site, consistent with its binding preference for RanGTP [72]. The proximity map was unable to categorise the shuttling proteins MEX67 and Ran, likely because non-structured FG-NUPs are poorly represented in our bait repertoire. Direct BioID combined with orthogonal assessment through expansion microscopy was required to confidently place these putative export factors. The derived localisations are summarised in Fig 3E and are consistent with a mechanistically divergent Ran-dependent mRNA export in trypanosomes.

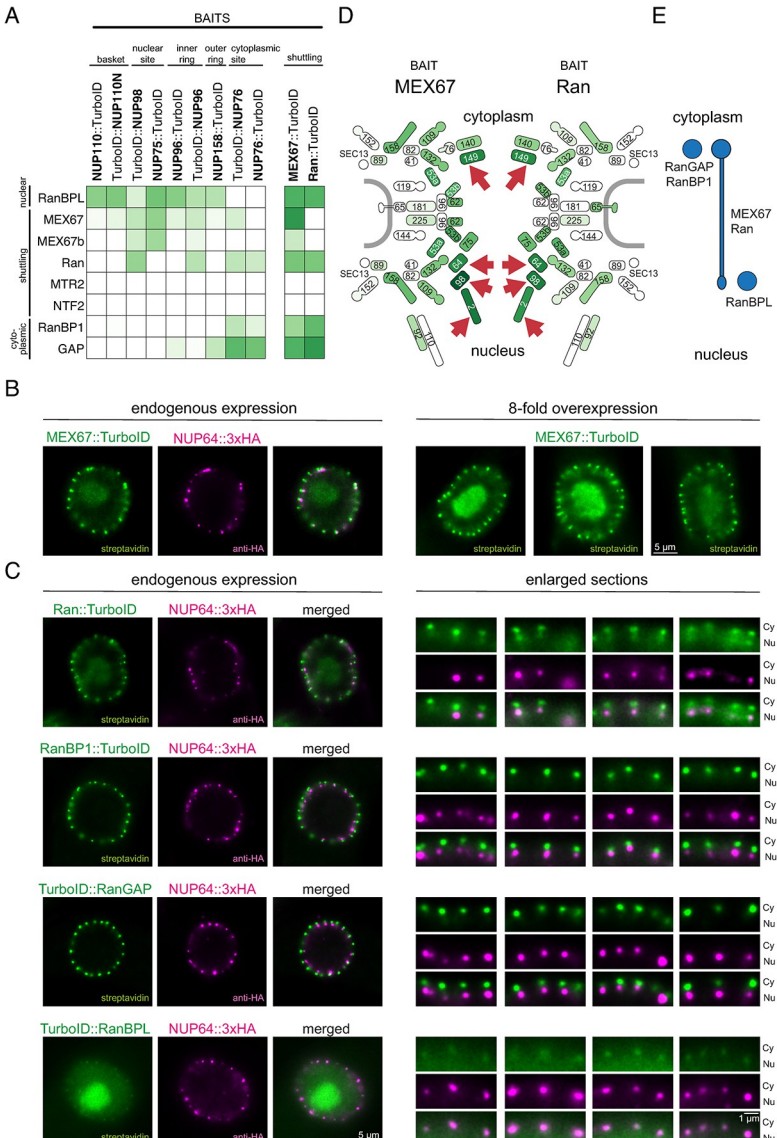

**Fig 3. Mapping the trypanosome mRNA export system to the nuclear pore. (A)** Mass spectrometry data (*t* test difference values) from proximity labelling experiments were used to map the Ran-based mRNA export system to the trypanosome pore. Details on the mass spectrometry data can be found in S1 Table. **(B, C)** Ultra-expansion microscopy. MEX67, Ran, RanBP1, RanGAP, and RanBPL were expressed as C-terminal (MEX67, Ran RanBP1, RanGAP) or N-terminal (RanBPL) fusions to TurboID in a cell line co-expressing the nuclear-site marker NUP64::3xHA, all from the endogenous locus. Cells were expanded and the proteins detected with streptavidin and anti-HA. Images of single nuclei are shown (single plane of deconvolved Z-stacks with 10 iterations for MEX67, Ran, RanBP1, and RanBPL and 20 iterations for RanGAP). For MEX67, the streptavidin signal was weak and 3 representative nuclei of an overexpression cell line are shown (B, right). For the other proteins, enlarged section of the nuclear envelope of the same or other nuclei are shown (C, right). **(D)** Proximity labelling of the nuclear pore by MEX67 and Ran. MEX67 and Ran were expressed as C-terminal TurboID fusions from the endogenous loci and biotinylated peptides were analysed by mass spectrometry. The labelling of NUPs by MEX67 (left) and Ran (right) is shown coloured based on their *t* test difference values in comparison to wild-type cells. Asymmetric NUPs are marked with a red arrow. **(E)** Schematic summary of our localisation data of the Ran-based mRNA export system. NUP, nucleoporin.

## Predicting the position of unknown proteins within the pore

To predict the localisation of the remaining 38 nuclear pore-localised proteins more accurately, we included the proximity labelling data of MEX67 and Ran to our proximity map.

Fifteen of the 38 nuclear pore-localised proteins are karyopherins (S7 Fig), five of which have not been previously classified as karyopherins but have unequivocal structural homology to importin and exportin-like folds predicted by FoldSeek (S7B Fig); these include a putative orthologue to the importin Hikeshi (Tb927.1.1400) that is specialised on the import of Hsp70-family proteins [79] (S7C Fig). Karyopherins were mostly not or poorly labelled by our proximity map (S7A Fig). The likely reason is their preferred interaction with FG-NUPs rather than structured NUPs, similar to what we observed for MEX67 and Ran (compare Fig 3A). Exceptions are XPO1 (exportin 1), known to be involved in the transport of both mRNAs and tRNAs [80], which is labelled by all bait proteins and 2 XPO-like proteins labelled by a subfraction of the baits.

An additional 5 proteins with nuclear pore localisation were not labelled by either of the bait proteins (S8 Fig). For three of these proteins, Tb927.11.1000, Tb927.10.12200, and Tb927.10.8160, the reason could be failed detection due to small size [55]. None of these small proteins has homologues outside of trypanosomatids and their function is unknown. Tb927.10.8160 has the strongest nuclear pore localisation ([35], S8B Fig) and high-throughput phenotyping indicates an essential function [81]. The 2 larger proteins (Tb927.1.3230 and Tb927.9.12700) do not have very prominent nuclear pore localisation ([35], S8B Fig). For Tb927.9.12700, biochemical data indicate glycosomal localisation [82] and Tb927.1.3230 could be the trypanosomatid ortholog of the ribosome biogenesis factor Rix7 [83]. Their lack of labelling might thus be due to poor or absent nuclear pore localisation.

The remaining 18 proteins were labelled by at least one of the bait proteins (Fig 4A). Strikingly, none were labelled by the cytoplasmic-site marker protein NUP76, suggesting absence of further proteins with exclusive cytoplasmic localisation, other than the NUP76 complex, RanGAP, and RanBP1. Moreover, not a single protein was exclusively labelled by the outer ring protein NUP158 [55], with the one exception of Tb927.11.13080. The (near) absence of combined labelling by NUP76 and NUP158 suggests that the outer ring proteome may be complete. Instead, these 18 proteins were either labelled by baits of the nuclear site or inner ring or both. We present the data as Pearson-distance clusters, with manual placements of proteins with insufficient labelling. Four proteins are not included to the clustering analysis: for one (SENP) the labelling pattern was too unique and 3 proteins were only labelled by MEX67 and/or Ran.

The majority of these 18 proteins is unique to Kinetoplastida or even to Trypanosomatida and lack functional annotations. Only 2 proteins have readily identifiable homologues outside Kinetoplastida: Tb927.10.9020 has homology to the non-catalytic, substrate binding subunit of the tRNA methyltransferase Trm6/Gcd10, responsible for adenosine(58)-N(1) methylation, a modification present in many eukaryotic tRNAs [84]. The second protein, Tb927.9.2220, is a SUMO protease of the Ulp/SENP (ubiquitin-like protease/sentrin-specific protease) family with potential function in resolving stalled DNA replication forks [85]. The remaining 16 proteins contain 5 proteins with predicted basket or inner ring localisation that were previously identified as lamina-associated proteins (LAPs), based on their interactions with the lamina-like proteins NUP1 and NUP2 [32]. Two of these LAPs, LAP71 and LAP102, are basket specific in our map, as expected for lamina associated proteins. Two further LAPs, LAP73 and LAP59, have exclusive inner ring prediction.

Of significant interest is basket/inner ring-predicted LAP173, which has a Sac3/GANP domain and was suggested to be the orthologue to Sac3 and sole representative of a potential

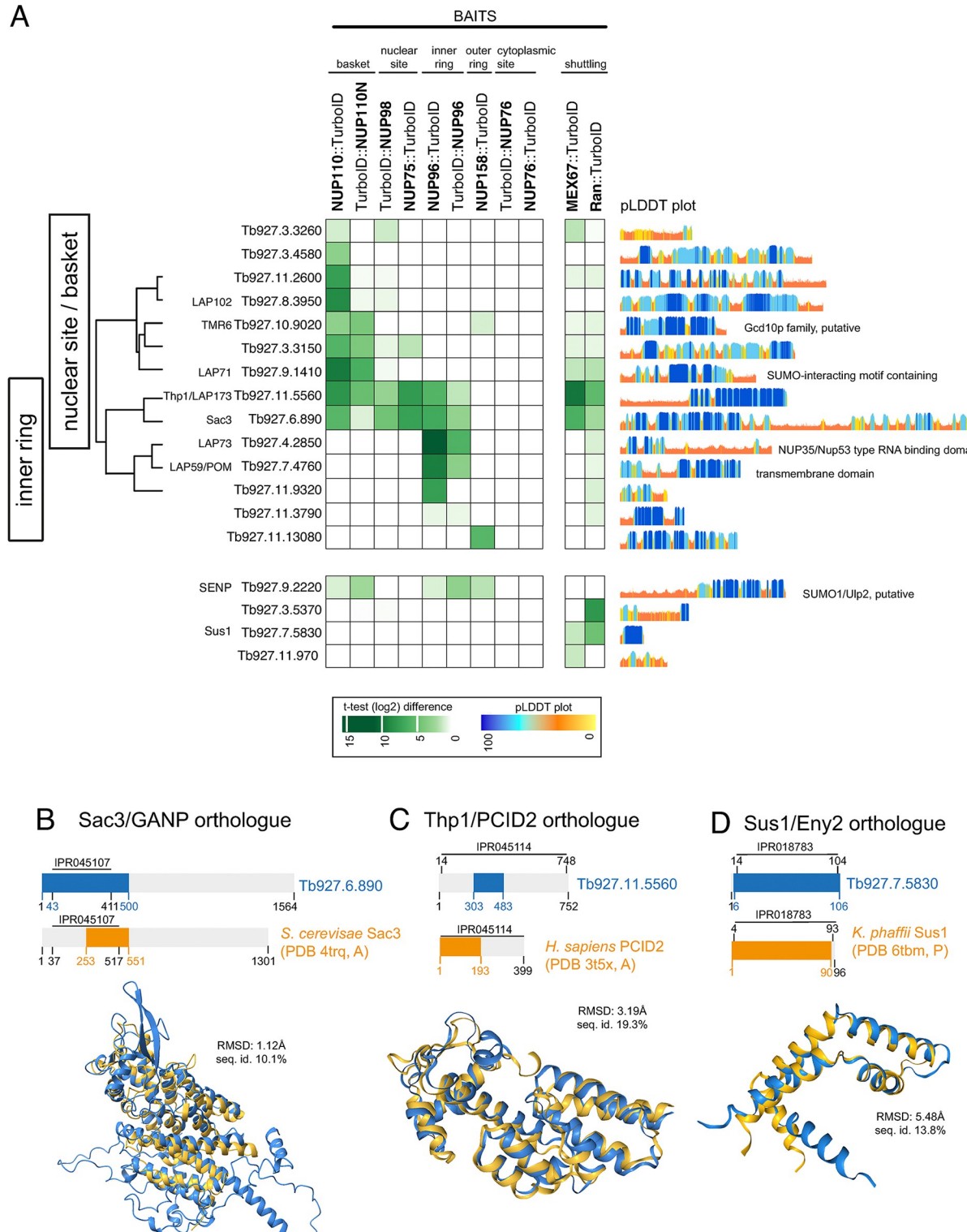

**Fig 4. Characterisation of unknown nuclear pore proteins and their localisation over the nuclear pore complex. (A)** Mass spectrometry data (*t* test difference values) from a range of proximity labelling experiments were screened for labelling of nuclear pore localised proteins that are not NUPs or karyopherins. All proteins that were labelled by at least one of the bait proteins are shown partially clustered using Pearson-correlation. Annotations are explained in the text. pLDDT plots from Trypanosomatid-optimised AlphaFold2 models [39] are shown on the right. Details on the mass spectrometry data can be found in S1 Table. **(B–D)** Structural alignments of AlphaFold2 models of the *T.*

*brucei* TREX-2 complex candidates Sac3 (B), Thp1 (C), and Sus1 (D) with PDB structures of the respective TREX-2 complex proteins from other organisms, using Foldseek [40]. The regions of the proteins that were used for the structural alignments are shown in the schematics in orange (*T. brucei*) and blue (other organisms). The root mean square deviation of atomic positions (RMSD) and the sequence identity are shown for the superimposed regions. NUP, nucleoporin; RMSD, root mean square deviation.

trypanosome TREX-2 complex [32]. Association with MEX67 was observed by affinity purifications [30] and BioID ([55], Fig 4A). In fact, the Sac3/GANP domain of a LAP173 model predicted by a trypanosome-optimised AlphaFold2 [39,41] displays remarkable structural homology to the equivalent region of an experimentally resolved *S. cerevisiae* Sac3 structure (RMSD 1.12Å, [86]), despite poor sequence conservation (Fig 4B).

Motivated by the presence of a putative Sac3, we used Foldseek [40] to search for structural homologues of the remaining TREX-2-complex components, using AlphaFold2 models as inputs [39,41]. We identified the Csn12-like domain containing protein Tb927.11.5560 as a putative Thp1 orthologue (Fig 4C), with structural homology to the human Tph1 homologue PCID2 (PDB entry 3T5X; TM score 0.82), while the primary sequence is, again, poorly conserved. Just like Sac3, our proximity map places this Thp1 candidate to both, nuclear site of the pore and inner ring. Moreover, we identified Tb927.7.5830 as a putative Sus1 orthologue with highest structural similarity to Sus1 of the yeast *K. phaffii* [87] (Fig 4D). The Sus1 candidate protein is not labelled by NUPs, presumably due to its small size. However, all 3 trypanosome TREX-2 complex candidates, including Sus1, are strongly labelled by MEX67, a prototypic Sac3 interactor in ophistokonts [88], supportive of a potential role in a trypanosome TREX-2 complex (Fig 4A).

In conclusion, our extended proximity map granted mapping the majority of nuclear pore localised proteins to a subregion of the pore. We found no evidence for the existence of any further proteins asymmetrically distributed to the cytoplasmic-site indicating the entirety of the cytoplasmic site-specific proteome of the pore is the NUP76 complex, RanGAP, and RanBP1, plus shuttling proteins. Instead, we predict a diverse cohort of proteins with preferential localisation to the basket or nuclear site of the pore, including 3 putative TREX-2 complex proteins with proximity to MEX67, indicative of a conserved function.

## The trypanosome NUP76 complex as a cytoplasmic mRNA remodelling hub

We detected the NUP76 complex (NUP76, NUP140, NUP149) exclusively at the cytoplasmic site (Fig 1A and 1E) and a previous study has shown the interaction of this complex with MEX67 under high stringency conditions [30]. In combination, these data suggest that the NUP76 complex is the trypanosomes cytoplasmic mRNP binding hub that serves as remodelling platform. In opisthokonts, the cytoplasmic mRNP remodelling platform is based on the heterotrimeric complex composed of Nup82/Nup159/Nsp1 in yeast and NUP88/NUP214/NUP62 in human (Fig 5A) [15,16]. The 3 proteins are connected via a C-terminal parallel coiled-coil structure. In yeast, both Nup82 and Nup159 possess N-terminal β-propellers that provide direct binding platforms for Nup145 (which recruits Gle2) and the RNA helicase Dpb5 (recruiting Gle1); in human, the complex is built in the same way from the respective human homologues (Fig 5A). Yeast Nsp1 and Nup159 (NUP62 and NUP214 in human) possess FG repeat regions. *T. brucei* NUP76 has been previously suggested as Nup82/NUP88 (yeast/human) homologue [30]; indeed, the AlphaFold2 model of NUP76 [39,41] shows an analogous structural organisation with a β-propeller at the N-terminus, interrupted by a long, disordered coil, and a three-helical coiled-coil at the C-terminus (Figs 5B, 5C, and S9). Moreover, *T. brucei* NUP76 may share its β-propeller interactions with Nup82/NUP88: orthologues

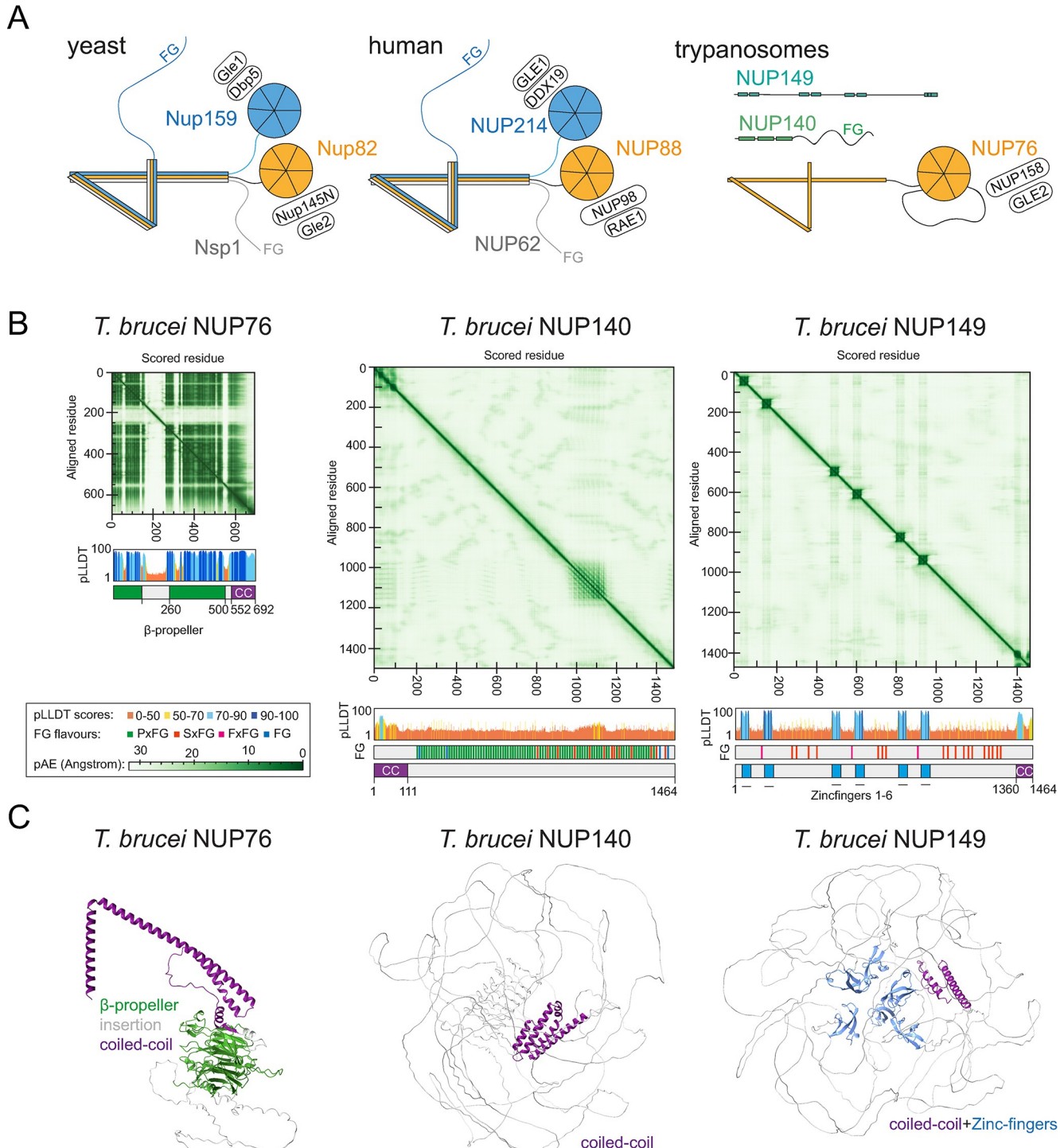

**Fig 5. The *T. brucei* NUP76 complex is only partially conserved. (A)** Schematics of the cytoplasmic filament complex from yeast and human (modified from [16], not to scale. The proteins of the trypanosome NUP76 complex are shown for comparison (left). Note that trypanosomes do have orthologues to NUP145N and Gle2, but it is not known whether these directly interact with NUP76. **(B)** pAE and pLDDT plots of trypanosomatid-optimised AlphaFold2 models of NUP76, NUP140, and NUP149. Each protein is also shown schematically with all predicted domains and, for NUP140 and NUP149, with positions and types of FG repeats. **(C)** Models of trypanosomatid-optimised AlphaFold2 predictions of NUP76, NUP140, and NUP149. Structured parts are coloured, disordered regions are shown in grey. NUP, nucleoporin.

to both Nup145N/NUP98 (yeast/human) and Gle2/RAE1 (yeast/human) can be readily identified in trypanosomes [29,30]. However, the 2 remaining TbNUP76 complex components, TbNUP140 and TbNUP149, do not exhibit detectable structural homology to the NUP82/NUP88 partner proteins Nup159/NUP214 (yeast/human) and Nsp1/NUP62 (yeast/human) (Fig 5A–5C) as based on AlphaFold2 predictions. TbNUP140 consists almost entirely of FG repeats of the PxFG type, apart from a small N-terminal stretch with a coiled-coil structure that is predicted with low confidence. NUP149 is not FG rich, with only few FG motifs of the SxFG and of the FxFG type but contains 6 zinc fingers sparsed by coils and potentially a small coiled-coil region at the C-terminus [30].

To investigate whether the trypanosome NUP76 complex is involved in mRNA export, we depleted the protein with an auxin-inducible degron system. Both alleles of the NUP76 gene were C-terminally fused to the OsAID-3xHA sequence, in a cell line that expressed the necessary components for the auxin degron system; the cell line was confirmed by diagnostic PCR (S11 Fig). Upon induction with the auxin derivative 5-Ph-IAA, the NUP76::OsAID-3xHA protein was depleted within 2 h (Fig 6A), followed by growth arrest (Fig 6B) and accumulation of poly(A) signal in the nucleus that was saturated 4 h post induction (Figs 6C and S12–S14). This phenotype is similar to the one observed upon Nup82 depletion in yeast [33,34], suggesting that NUP76 is the functional orthologue to yeast Nup82 with a crucial role in mRNA export. In order to limit the possibility that the observed blockade of mRNA export is an indirect effect, i.e., the result of a disrupted pore architecture, we expressed a range of NUPs as N- or C-terminal eYFP fusions in the NUP76 auxin degron cell line to test whether their localisation to the pore is dependent on NUP76 (Figs 6D, 6E and S15). The localisation of the inner ring NUP96 was not affected by NUP76 depletion, suggesting that the overall pore structure remains intact (Fig 6D). Of the (putative) NUP76-associated proteins, only the pore localisation of NUP140 was clearly abrogated upon NUP76 depletion, while NUP149 and Gle2 still localised to the pore (serving as additional controls for pore integrity not being affected). Note that a slightly diminished pore localisation was observed for all 4 proteins, possibly caused by the disrupted mRNA export and general loss in fitness rather than a specific impact on nuclear pore architecture. Thus, NUP140 localisation to the pore is fully dependent on NUP76, while NUP149 and Gle2 appear to be anchored independent of NUP76.

In conclusion, the cytoplasmic site-localised NUP76 complex of trypanosomes, consisting of NUP76, NUP140, and NUP149, is distinct from the cytoplasmic mRNA remodelling complex of yeast and human. While NUP76 is the likely functional homologue to Nup82/NUP88 from yeast/human, NUP140 and NUP149 are trypanosome-unique with no sequence or structural homology to cytoplasmic site (filament) proteins from opisthokonts. The absence of a Nup159/NUP214 (yeast/human) orthologue in trypanosomes correlates with the absence of orthologues to its interaction partners Dbp5/DDX19 and Gle1/GLE1 (yeast/human), suggestive of significant mechanistic differences on the mRNA remodelling mechanisms in trypanosomes.

## Discussion

The compartmentalisation of hereditary information in the nucleus necessitated the invention of a gateway allowing mRNPs and a variety of other essential cargos to cross the nuclear envelope. The study of nuclear pores in evolutionary divergent eukaryotes, such as the ancient trypanosomes, is fundamental to understand the evolutionary origin of the nucleus and decipher the complexity of the nuclear pore as platform with multiple cargo routes. Our study contributes a roadmap of the trypanosome nuclear pore, reporting conserved and non-conserved features and devising a plethora of new leads for further exploration.

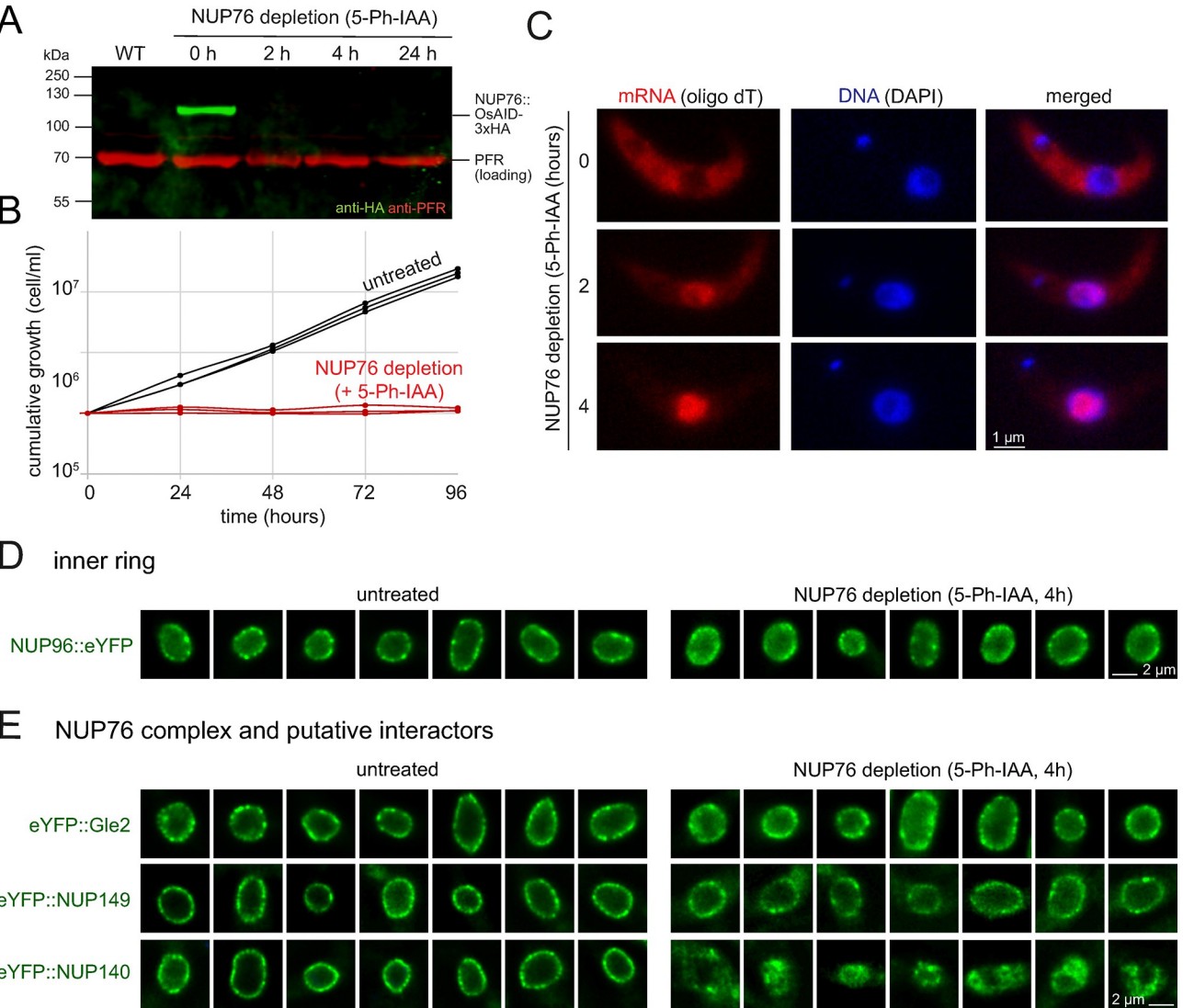

**Fig 6. Depletion of TbNUP76 causes nuclear poly(A) accumulation and loss of NUP140 pore localisation.** NUP76 protein was depleted using a degron system based on induction with the auxin derivative 5-Ph-IAA. Both alleles of NUP76 were replaced by NUP76 fused to OsAID-3xHA at the C-terminus. **(A)** The depletion of the NUP76 protein at 2, 4, and 24 h of induction was monitored on a western blot using anti-HA to detect NUP76 and anti-PFR as loading control. Wild-type (WT) cells served as negative controls. Data of one representative clonal cell line are shown. **(B)** Growth was monitored over 5 days following NUP76 depletion. Data of 3 independent clonal cell lines are shown. Raw data can be found in S3 Table. **(C)** In situ hybridisation: cells were probed with oligo dT to monitor mRNA localisation. The DNA is labelled with DAPI. One representative cell is shown for untreated cells and cells after 2 and 4 h of NUP76 depletion (method: sum slices of 75 images recorded at 140 nm distance). Fluorescent profiles and images with more cells are shown in S12–S14 Figs. Representative data of one out of 3 clones is shown. **(D, E)** An N-terminal eYFP fusion of NUP96 and C-terminal eYFP fusions of NUP140, Nup149, and Gle2 were expressed from endogenous loci in the NUP76 degron cell line. The eYFP fluorescence of 7 randomly selected nuclei is shown before and after induction of NUP76 depletion. Additional nuclei are shown in S15 Fig. NUP, nucleoporin.

This study is an update of the trypanosome nuclear pore composition and is based on the foundations set by earlier studies: In 2009 [29] had identified all trypanosomes NUPs using predicted structural similarities in pre-AlphaFold times. In 2016, Obado and colleagues have used a combination of high stringency precipitation of cryomilled cell material and electron microscopy with immunogold labelling to present the first detailed model of the trypanosome pore [30]. Finally, the genome-wide localisation database TrypTag has provided a

comprehensive list of proteins with pore localisation [35]. We took advantage of all these available data to create a revised map of the pore, by combining expansion microscopy with a novel way to globally analyse proximity labelling data. Mostly, our map is in agreement with the earlier map. We did one major correction, which is the placement of the NUP76 complex and the NUP64/NUP98 complex to the cytoplasmic and nuclear site, respectively. The previous symmetric assignment of these complexes was solely based on electron-microscopy detection of immunogold labelled NUP-GFP fusions. This method may be error prone due to large distance of the gold particle to the target protein (GFP+antibody tandem tag) and, even more importantly, it is very difficult to accurately determine the centre of the pore, as membranes are poorly visible. In contrast, in expansion microscopy, the localisation is determined in relation to another pore marker protein and does not depend on membrane detection, and, at least in UExM, the localisation error caused by the antibodies and tag is much smaller because the labelling is done after the expansion. We do therefore believe that expansion microscopy reflects the localisation of pore proteins more accurately. Nevertheless, the conflicting results prompted us to confirm our findings with orthogonal methods and we chose a heat map, generated from proximity labelling mass spectrometry data of marker proteins at different positions within the pore. The heat map agreed with all expansion microscopy data, and, importantly, it does not rely on the proteins being modified with a tag, which could affect localisation. Both the heat map and the expansion microscopy have their pitfalls and can potentially create wrong results. However, the combination of the methods increases the confidence, in particular when further combined by targeting multiple subunits of one complex rather than just one, as we did for example for the NUP76 complex.

## NUP76: A partially conserved cytoplasmic site-specific complex with connections to mRNA export

Of the NUPs, only the NUP76 complex (NUP76, NUP140, and NUP149) appears localised specifically to the cytoplasmic site. The NUP76 complex had been previously suggested to be part of the mRNA remodelling platform, as all 3 proteins co-isolated with Mex67 under high stringency conditions [30]. Consistent with this hypothesis, we now show exclusive cytoplasmic localisation of the NUP76 complex (Fig 1) as well as significant structural similarities of NUP76 with the scaffold mRNA remodellers of yeast and human, NUP82 and NUP88, respectively (Figs 5 and S9) and nuclear mRNA accumulation upon NUP76 depletion (Fig 6C). While NUP76 is a likely homologue to yeast NUP82 or human NUP88, the other proteins of the NUP76 complex, NUP140 and NUP149, appear unique to trypanosomes and share no similarity with the proteins that interact with yeast Nup82 or human NUP88. They possess no predictable structured elements, with the exception of NUP149 which possess four zinc finger motifs. Notably, zinc fingers are also present in the human cytoplasmic-filament NUP358 and the nuclear-site localised NUP153, both absent from trypanosomes, where they engage in Ran binding [89,90]. However, the zinc fingers of TbNUP149, confidently predicted as 3 β-hairpin strands with 4 cysteines side chains coordinating a zinc ion by AlphaFold2 and AlphaFold3 [91] appear to lack obvious sequence or structural homology to the zinc fingers of human NUP358 and NUP153 (S10 Fig) and whether they nevertheless promote Ran binding remains to be investigated. Notably, NUP149 is heavily biotinylated by Ran-TurboID, indicative of a possible interaction (Fig 3B).

Even though the NUP76 complex in trypanosomes is different to the complexes from yeast and human, one similarity is worth mentioning: NUP76 depletion disrupts pore localisation of NUP140, but not of NUP149, just like NUP88 depletion in human disrupts pore localisation of NUP214, but not Nup62 [92].

## A mechanistically divergent Ran-dependent mRNA export pathway in trypanosomes

Apart from the NUP76 complex, we mapped RanGAP and RanBP1 to the cytoplasmic site of the nuclear pore. Their sole cytoplasmic localisation is suggestive of a conserved function of these proteins in triggering GTP hydrolysis of RanGTP and thus disassembly of exportin-cargo-RanGTP and importin-RanGTP complexes.

Trypanosome RanGAP is phylogenetically more closely related to a RabGAP [93] but its proposed function as RanGAP [30] is further corroborated by our study. Pore-anchoring of RanGAP and underlying mechanisms significantly vary across species, ranging from a SUMO-dependent interaction with the metazoan-specific RanBP2/NUP358 [75,94–96], over a WPP domain-specific to plant RanGAP that interacts with a plant specific nucleoporin [97,98], to no pore localisation at all in *S. cerevisiae* and *S. pombe* [99,100]. The mechanism of pore localisation of trypanosome RanGAP is thus likely unique and may involve interactions with likewise unique trypanosome specific NUPs such as NUP140 and/or NUP149. Trypanosome RanBP1 consists of a disordered 30-amino acid stretch followed by a conserved RanBP domain (S16A Fig) and it remains unclear whether it has binding sites to the pore or simply concentrates in sites of cargo docking.

The lack of a Dbp5 homologue and the association between MEX67 and Ran implies that trypanosomes employ the Ran-GTP gradient for mRNA export [30]. In fact, while Ran is predominantly nuclear localised in humans [101], in trypanosomes we find biotinylated Ran targets on both sites of the pore, possibly reflecting Ran engagement in cargo import and export [101]. This unique dual usage of the Ran pathway for both mRNA and protein cargo presents a formidable challenge for export/import ratio moderation. It is tempting to speculate that RanGTP is anchored at the basket site awaiting the MEX67 bound mRNP, which is then liberated on the cytoplasmic site driven by RanGAP-catalysed GTP hydrolysis. The lack of Dbp5 suggests that ATP-dependent mRNP disassembly at the cytoplasmic site of the pore is dispensable in trypanosomes, implying a fundamentally different mode of interaction between MEX67 and mRNA. Indeed, trypanosome MEX67 uniquely carries a CCCH-type zinc finger instead of the canonical RNA recognition motif containing RNA binding domain (RRM/RBD) found in ophistokonts [102,103], and trypanosomes lack mRNA adaptors (SR proteins) that would require stripping during cytoplasmic remodelling. Thus, the exported trypanosome RNP may exhibit lower stability and complexity, making a remodelling RNA helicase redundant.

We have identified another potential component of the Ran system at the nuclear site of the pore: RanBPL has a Ran-binding domain which is very similar to the one of cytoplasmic-site localised RanBP1 but has a longer disordered N-terminal stretch (S16 Fig) and was previously characterised as Ran-binding protein with a clear preference to RanGTP over RanGDP [72]. Thus, RanBPL may be the trypanosome functional counterpart to basket proteins Nup2/NUP50 (yeast/human), which also possess Ran-binding domains. While the multiple roles of Nup2/NUP50 remain largely elusive, one known function is the acceleration of protein import complex disassembly through stimulation of RanGEF/RCC1 activity [104], analogous to the function of RanBP1 as enhancer of RanGAP activity at the cytoplasmic site [105]. In trypanosomes, a RanGEF has not yet been identified and the absence of a detectable RanGEF/RCC1 domain among the proteins biotinylated by Ran indicates that a trypanosome RanGEF/RCC1 is either absent or divergent. Theoretically, RanBPL has the potential to compensate for the absence of the canonical RanGEF/RCC1: instead of directly catalysing the GDP to GTP exchange, RanBPL1 could act by stabilising RanGTP and preventing GTP hydrolysis, driving the equilibrium towards RanGTP. However, the observation that RanBPL silencing evokes only a mild growth phenotype (Brasseur and colleagues [72]) argues against this hypothesis.

Altogether, our study fortifies the hypothesis of a Ran-dependent mRNA export pathway in trypanosomes and opens new avenues for exploration of the underlying molecular mechanisms. Of potential interest in this context is also the hypothetical protein Tb927.3.5370 that is strongly labelled by Ran, but not by any NUPs.

## Newly identified proteins with (predicted) localisation to the nuclear site of the pore

While only 5 proteins are specific to the cytoplasmic site, the nuclear site of the pore appears more complex. Next to the previously described basket proteins NUP110 and NUP92 [29,30] and RanBPL (discussed above), we found exclusive nuclear site localisation for the FG nucleoporins NUP64 and NUP98, putative TREX-2-complex proteins and up to 8 further proteins (the number is depending on how threshold is defined) that are mostly unique to trypanosomes. The FG-NUPs NUP64 and NUP98 are unique to trypanosomes and in a complex with NUP75 [30] which appears to extend to the inner ring via NUP75. NUP64 and NUP98 were previously suggested to be the (functional) orthologues of *S. cerevisiae* Nup1 and Nup60, as they carry the same FG-type and engage in an interaction with the putative Sac3 homologue [30,32]. Our data now show the exclusive nuclear-side position of these NUPs, in full support of this model.

The TREX-2 complex was believed to be absent from trypanosomes, with the possible exception of Sac3 [32] (Fig 4B). We have now identified proteins within the cohort of nuclear pore-localised proteins [35] that show structural similarity to Thp1 and Sus1 (Fig 4C and 4D). Moreover, the orthologues to Thp1 and Sac3 have a predicted localisation at the nuclear site of the pore. All 3 trypanosome candidate TREX-2 components now await experimental analysis to understand the mechanistic details of the trypanosome mRNA export platform. The 2 further TREX-2 components, Sem1 and Cdc31 [14], were not identified within the trypanosome nuclear pore-localised proteins. These are either absent from the trypanosome TREX-2 complex, or failed identification either due to poor structural conversation or because the proteins were not identified as pore-localised by TrypTag [35].

Six of the 8 further proteins with predicted localisation at the nuclear site of the pore are trypanosome-unique with no obvious homologies and further experiments are essential to uncover their functions. Two of the proteins have predicted functions: Tb927.10.9020 is the likely homologue to the non-catalytic subunit of the tRNA methyltransferase TRM6 and was predicted as a basket-specific nuclear pore protein in our heatmap, with minor labelling by the outer ring NUP158 and by MEX67 and Ran (Fig 4A). The protein exhibits strong nuclear pore localisation [35], which is in contrast to the nuclear localisation observed for *S. cerevisiae* and *A. thaliana* Gcd10/TmR6 [106,107]. While localisation to the nuclear pore and/or envelope is not unheard of for tRNA modifying enzymes [108,109], this finding requires further investigation, as, conversely, the putative *T. brucei* homolog of the corresponding catalytic subunit, TRM61/Gdc14 (Tb927.11.11660), localises to the nuclear lumen/nucleoplasm [35]. The other protein with predicted basket localisation, Tb927.9.2220, has homologies to an ubiquitin-like protease/sentrin-specific protease (Ulp/SENP) that may function in resolving stalled DNA replication forks [85]. Both Ulp1 of yeast and SENP2 of human have nuclear pore localisation [110,111] and the latter was localised to the nuclear site of the pore, consistent with our map [110]. TbSENP has a rather unique biotinylation pattern that did not clustered with the biotinylation pattern of any other nuclear pore protein: it is labelled by the basket-localised NUP110 and by NUPs of the inner and outer rings, but neither by the nuclear site-specific NUP98 and NUP75 nor by the cytoplasmic site-specific NUP76. The reason for this unusual labelling pattern remains unknown and requires further investigation.

### Proteins with inner-ring prediction

Two non-NUP proteins have exclusive inner ring prediction: LAP59 and LAP73. LAP59 was previously co-isolated with the lamina-like proteins NUP1 and NUP2, is conserved across eukaryotes and the presence of 2 N-terminal transmembrane domains suggests it to be a pore membrane protein (POM) [32]. LAP73 has a divergent NUP35/Nup53 type RNA-binding domain [32] and, interestingly, the *T. brucei* orthologue to Nup53, TbNup65, is anchored to the nuclear envelope via a trans-membrane helix [30]. This raises the possibility of a nuclear envelope and thus inner ring localisation of LAP73 via binding to TbNUP65. However, immunoprecipitation assays failed to establish an inner ring association with LAP73 [30], albeit it is possible that the interaction is weak and thus exclusively detectable in BioID.

### Conclusions

Our revisited map of the trypanosome nuclear pore conforms to the pattern of conservation at the core scaffold regions and diversity at the borders of the pore [31]. We discovered an asymmetric architecture, confidently placing the NUP76 complex exclusively to the cytoplasmic site and defining the sole localisation of the trypanosomatid-exclusive FG NUPs NUP64 and NUP98, at the basket site. Notably, this corrects the current view of a largely symmetric trypanosome nuclear pore and ultimately supports moderation of directional nucleocytoplasmic transport which is crucially dependent on asymmetric components at the nuclear pore borders in other systems. For the NUP76 complex, our data strongly indicates a crucial function as cytoplasmic mRNP remodelling hub, analogous to the Nup82/NUP88 complex in opisthokonts, while the presence of trypanosome-unique NUP140 and NUP149 implies significant mechanistic difference. Mapping of the export factors Mex67 and Ran elucidated further divergence, supporting a trypanosome-specific, Ran-dependent export system. Lastly, we present a comprehensive assignment of pore localised proteins to subregions of the nuclear pore that resulted in the identification of novel nuclear pore components, including 3 putative members of a trypanosome TREX-2 complex. Altogether, our approach delivers asymmetric and novel nuclear pore components, including positional information, which can now be interrogated for functional roles to explore trypanosome-specific adaptions of nuclear transport, export control, and mRNP remodelling.

### Supporting information

**S1 Fig. Establishment and validation of the expansion microscopy protocols.**
(PDF)

**S2 Fig. Additional proExM images of the NUP76 complex proteins.**
(PDF)

**S3 Fig. UExM images of NUP64-4xTy1 with NUP76 complex proteins tagged with 3xHA (labelled using antibodies).**
(PDF)

**S4 Fig. Statistical analysis of TurboID experiments.**
(PDF)

**S5 Fig. Inducible overexpression of MEX67 (growth curves and western blotting to check for expression levels).**
(PDF)

**S6 Fig. Nucleolus can be identified by reduction in DAPI stain.**
(PDF)

**S7 Fig. Mapping proteins with the proximity map: Karyopherins.**
(PDF)

**S8 Fig. Mapping proteins with the proximity map: unlabelled proteins.**
(PDF)

**S9 Fig. Structures of yeast NUP82, human NUP88 and predicted *T. brucei* NUP76.**
(PDF)

**S10 Fig. The NUP149 zinc fingers in comparison to the ones from human NUP153 and NUP358.**
(PDF)

**S11 Fig. Verification of NUP76 auxin degron cell lines by diagnostic PCR.**
(PDF)

**S12 Fig.** Additional poly(A) FISH images and fluorescence profiles of NUP76 depleted cells.
(PDF)

**S13 Fig.** Additional poly(A) FISH images and fluorescence profiles of NUP76 depleted cells.
(PDF)

**S14 Fig.** Additional poly(A) FISH images and fluorescence profiles of NUP76 depleted cells.
(PDF)

**S15 Fig. Effect of NUP76 depletion on NUP140, NUP149, Gle2, and NUP96 localisation: additional images.**
(PDF)

**S16 Fig. AlphaFold2 models of RanBP1 and RanBPL.**
(PDF)

**S1 Raw Images. Contains raw images of all blots.**
(PDF)

**S1 Table. Oligo sequences.**
(XLSX)

**S2 Table. Mass spectrometry data.**
(XLSX)

**S3 Table. Raw data for all graphs.**
(XLSX)

**S4 Table. Raw data of the FISH profiles (S12–S14 Figs).**
(XLSX)

## Acknowledgments

We are grateful to the OMICS Proteomics BIOCEV core facility for excellent technical service. We like to thank Eva Kowalinski (EMBL, Grenoble, France) for expert help with Alphafold and Colabfold. We are grateful to Mark Carrington for providing the highly useful auxin degron system.

## Author Contributions

**Conceptualization:** Bernardo Papini Gabiatti, Martin Zoltner, Susanne Kramer.

**Data curation:** Bernardo Papini Gabiatti, Johanna Krenzer, Silke Braune, Martin Zoltner.

**Formal analysis:** Johanna Krenzer, Martin Zoltner.

**Funding acquisition:** Martin Zoltner, Susanne Kramer.

**Investigation:** Bernardo Papini Gabiatti, Johanna Krenzer, Silke Braune, Martin Zoltner.

**Methodology:** Bernardo Papini Gabiatti, Johanna Krenzer, Silke Braune, Timothy Krüger, Martin Zoltner.

**Project administration:** Martin Zoltner, Susanne Kramer.

**Resources:** Martin Zoltner.

**Supervision:** Martin Zoltner, Susanne Kramer.

**Visualization:** Susanne Kramer.

**Writing – original draft:** Bernardo Papini Gabiatti, Martin Zoltner, Susanne Kramer.

**Writing – review & editing:** Bernardo Papini Gabiatti, Martin Zoltner, Susanne Kramer.

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
