## [Editor Report · Decision Letter 0]

25 Oct 2024

Dear Dr Kramer, 

Thank you for submitting your manuscript entitled "A combination of expansion microscopy and proximity labelling reveals conserved and unique asymmetric functional hubs at the trypanosome nuclear pore" for consideration as a Research Article by PLOS Biology. I would like to apologize for the long delay in coming back with a response. 

Your manuscript has now been evaluated by the PLOS Biology editorial staff, as well as by an academic editor with relevant expertise, and I am writing to let you know that we would like to send your submission out for external peer review.

Once your full submission is complete, your paper will undergo a series of checks in preparation for peer review. After your manuscript has passed the checks it will be sent out for review. To provide the metadata for your submission, please Login to Editorial Manager (https://www.editorialmanager.com/pbiology) within two working days, i.e. by Oct 27 2024 11:59PM.

Kind regards,

Melissa

Melissa Vazquez Hernandez, Ph.D.

Associate Editor

PLOS Biology

---

## [Decision Letter · Decision Letter 1]

5 Dec 2024

Dear Dr Kramer,

Thank you for your patience while your manuscript "A combination of expansion microscopy and proximity labelling reveals conserved and unique asymmetric functional hubs at the trypanosome nuclear pore" went through peer-review at PLOS Biology. Your manuscript has now been evaluated by the PLOS Biology editors, an Academic Editor with relevant expertise, and by several independent reviewers.

As you will see in the reviewer reports below, all reviewers provided positive feedback on the study, though they have highlighted a few areas for improvement. Specifically, Reviewer #1 requests the inclusion of quantitative microscopy results to better demonstrate mRNA localization, while Reviewer #3 suggests evaluating whether the localization of other pore components remains unchanged. Additionally, the reviewers recommend introducing a separate "Discussion" section, which we fully support. While no additional experiments are required, addressing these suggestions could significantly strengthen the manuscript. However, the study will require some revision and rewriting to align with the reviewers' feedback before it can be considered for further publication. 

**IMPORTANT - SUBMITTING YOUR REVISION**

*Resubmission Checklist*

*Published Peer Review*

*PLOS Data Policy*

*Blot and Gel Data Policy*

Sincerely,

Melissa

Melissa Vazquez Hernandez, Ph.D.

Associate Editor

PLOS Biology

REVIEWERS' COMMENTS:

Reviewer #1: 

This study represents a notable step forward in understanding the molecular architecture of the nuclear pore in Trypanosoma brucei parasites, valuable for the basic understanding of nuclear pore evolution as trypanosomes are among the earliest branching lineages of eukaryotes.

The key experimental data are expansion microscopy analysis of nuclear pore protein colocalisation for a subset of nuclear pore proteins, a proximity labelling (BirA/BioID-based) analysis of proximity of proteins within the nuclear pore and an elegant application of the auxin degron system to show that NUP76 is critical for mRNA export.

The experimental work appears overall of good quality. The expansion microscopy, BioID and auxin degron data appear to be of high quality with good controls, and I can see that the major results are supported and the major conclusions broadly justified.

Some of the microscopy results are presented qualitatively, where a basic image analysis would allow rigorous statistical support. Notably the localisation of mRNA relative to the nucleus in Figure 6C, showing nuclear accumulation of mRNA on NUP76 depletion. As this is a key result, a quantitative analysis of nuclear vs. cytoplasmic signal in a good number of cells and preferably from multiple biological replicates would be better. Especially, as 3 clones were generated, but data from only one is shown.

Figure 6D,E would similarly benefit from a quantitative analysis, although less obvious what that would be, and well mitigated by showing multiple example nuclei and the extended presentation in the paired supplemental figure.

Quantitative microscopy analysis is applied inconsistently, ie. cell lines listed in Figure 1A have measured signal separation presented in 1B, while those in 1C-E and 3B-C do not. Only HA-TY colocalisation appears to have been analysed. Is there a reason why HA-streptavidin should not also be measured? I think it would add value.

However, the text has some significant issues.

There is no conventional discussion section, and the conclusions section does not do a balanced discussion/summary of the result. Prevalence of discussion material in the results generally becomes more problematic as one progresses through the results section. There are numerous paragraphs which are entirely discussion, with no references to figures or data. As a particularly bad example, there are almost two pages of text in the results section with no reference to figures (lines 521 to 570). I got to this section and genuinely wondered if I had missed a "discussion" title.

However, the introduction, with comprehensive and clear referencing of previous work, is clear. The verbose and clear methods section is good, similarly figure legends are broadly clear.

The paper would greatly benefit from a clear conventional discussion section and moving of the long discussion-like paragraphs from the results to this new discussion section. Much of the extended discussion within the results seems to be about presence/absence of orthologs, structural features, and names of sequence/structure/functionally orthologous proteins in other species. Given a key impact of this work is understanding the evolution of the nuclear pore, summarising this information in a carefully structured table or matrix would be more informative than the very verbose text.

I wasn't able to find some key discussion points which seem to arise from this work, notably:

What properties of the previous immuno EM work might have spuriously given a symmetrical nuclear pore result, and an honest exploration of what limitations of the expansion microscopy/TurboID could exist which may have spuriously given an asymmetric nuclear pore result. My opinion is that the expansion microscopy data is likely to be more robust, however a full balanced discussion would bolster the claim of asymmetry in the trypanosome nuclear pore structure.

This is now the third or fourth complementary approach to generating a comprehensive "nuclear poreome" (following the intial systematic affinity purification, the genome wide tagging/TrypTag and the T. brucei LOPIT analysis). I realise there are comparisons made in the text, and one supplemental figure for comparison to tagging data, but a summary table would be very beneficial. Similarly, discussion about the potential methodological origins of inconsistencies.

There is a consistent problem with indicating when proteins are mutants (ie. when tagged with an epitope tag or proximity labelling tag):

Proteins should be referred to in the text and figures with clear indication of the tag and tagged terminus, eg. <protein>::GFP and GFP::<protein> for C and N terminal tagging respectively, whenever data is being derived from a mutant protein. Some figures make this relatively clear, for example Figure 3, while others, for example Figure 1, and the text generally do not.

Tagging terminus and selection of tag can have important biological consequences:

Protein tagging may cause an abnormal localisaiton, and it is important that conclusions drawn clearly indicate the evidence - localisation based on a small epitope tag may, for instance, be viewed as more trustworthy than a larger globular proximity labelling protein. Ideally, conclusions should be stated using this information - eg. based labelling by X::TurboID, Y is a ...

For proximity labelling experiments, lack of clear indication of which protein has been TurboID tagged leads to the text being confusing about what is doing the labelling and what is being labelled / which is bait and which is prey (see also below).

A key conclusion is that NUP76 is specific to the cytoplasmic side, there is evidence from HA-tagged NUP76 relative to TY tagged NUP110, and TurboID-tagged NUP76 relative to HA tagged NUP64. However, which terminus of NUP76 was tagged in these instances and, if it was the same one, could that have perturbed its localisation to be on the cytoplasmic side? This affects subsequent interpretation of the prey labelled by TurboID-tagged NUP76.

My balanced opinion is that it is unlikely that the tagging will have substantially affected NUP76 localisation, but the mutants proteins used to make the conclusions should be more clearly stated.

I noted one key concern about interpretation of the data: 

There is an important caveat that expansion microscopy (indeed, any microscopy) identifies the location of the small globular fluorescent/TurboID or epitope tag - not the location of the tagged protein.

At the scales being considered, proteins could have a large physical size such that the selected tagged terminus may substantially affect the observed localisation - even in the absence of functional perturbation by presence of the tag.

The proximity data mostly (but not always) uses both N and C terminal fusion with TurboID. Was the motivation here to understand the potentially different location of the N and C terminus? Why not a similar level of care for the microscopy?

When comparing to the existing immuno EM data, carefully and critically consider whether antibodies were directly to the target nuclear pore protein, or if they were to an epitope or fluorescent protein tag and, if so, at which terminus the tag was added.

Problematic terminology.

Please be critical in selecting terms, I found several confusing:

In a paper extensively using microscopy, avoid using the term "focused" outside of referring to the focal plane of a microscope. For example line 320 it is unclear whether you are focusing the microscope or your attention.

Avoid the term "nuclear" unless absolutely necessary, given this is a paper about nuclei. For example line 363, I'm left very confused about what "nuclear site" means in this context.

Saying a protein localises to a "single dot" is very unclear when talking about nuclear pores. For example line 450, RanGAP does not resolve as a single dot - in Figure 3C there are ~21 dots, plus a nucleolar signal. Please carefully check and rephrase all of these statements.

"bait" is commonly used in the proximity labelling literature but, without clear introduction, it is confusing. Especially as "bait" is only meaningful when paired with the "prey" term, which appears not to be used at all. In general, the language here is imprecise, for example 399, bait proteins to not do the labeling, TurboID does the labelling.

I realise nuclear pore "spokes" is a term used in the literature, but personally I find this confusing as radial spokes are rings, not spoked disks.

Consistently use N and C or amino and carboxy for tagging, mixed terminology is confusing.

Avoid qualitative terms like "many", and state the exact number, preferably as a fraction of the total (eg. in the abstract "many further asymmetrically localised nuclear pore components" -> "X out of Y nuclear pore proteins, Z of which were novel, were asymmetrically localised")

Specific minor points:

Line 29. Please be clear about radial vs. inside-outside symmetry, as written, this is ambiguous.

Line 30. Expansion microscopy and TurboID are well-established tools, I'm unclear what is novel here.

Line 38. This is presumably _predicted_ structural homology? Throughout the paper, make it clear when structures are predicted vs experimentally known.

Line 44. Typo for -> fort

Line 59. Unclear if Nup84/NUP107 means alternative names, or the names for the yeast and human protein respectively. Similar issues arise later.

Line 99. This introduction paragraph would benefit from a clear closing question that is addressed by your study.

Line 102. "A range or organisms" and "examples" is very uninformative here, as a key impact of your study is the ability to look at the specific evolution of the nuclear pore. A clearer summary of which species, perhaps specifically listing those outside of opisthokonts, would be more informative.

Line 109. "Puzzle-stone" isn't a common idiom, at least not one I know.

Line 169. Good practice to indicate the data version, or an indication of when sequences were gathered.

Line 247. Should indicate that this is PBS supplemented with Tween, BSA.

Line 280. This contains some jargon I am not familiar with at all, specifically "reverse and hits only identified by site" should be explained properly.

Line 306. Streptavidin imaging is not detection of only autobiotinylation, you also detect all proximal biotinylation, and expect the TurboID to be at the centre of that cloud.

Lines 209 to 316. This is not results text, should be discussion(?), or explained with the relevant data.

Line 325. Use consistent terminology for N and C vs. amino and carboxy.

Line 327. The image does not show a single dot, this is unclear. Need to explain the observation more accurately - ie. well-separated points around the periphery of the nucleus in the focal plane.

Line 332. Perhaps I missed it, but is the expansion factor experimentally determined or are you taking this as an assumption into the distance measurement calculation? Needs to be explained/acknowledged as a limitation.

Line 363. "nuclear site" is confusing terminology.

Line 368. This is incorrect, you do not resolve the rings at all. There are paired points, around the periphery of the nucleus, which can be interpreted as the inner and outer ring. At no point do you resolve the ring per se. Needs to be explained more clearly.

Line 388. "bona fide" is not a rigorous term. State the actual scientific logic, are these orthologs of NUPs from other species? Are they NUPs known from prior affinity purification work? etc.

Line 392. Alphafold does not predict structure or unstructure, it predicts structure with a measure of the accuracy of the predicted structure. Various studies have shown that low structure confidence correlates with regions predicted as unstructured by other dedicated unstructured domain prediction tools. This should be explained more clearly.

Line 395. "that we had placed asymmetrically localised to the nuclear site" is confusing. You haven't placed anything into this system, so why use the word 'placed'? Do you mean you found it localised to somewhere? What does 'nuclear site' mean? Presumably not the nucleus, do you mean the centre of the nuclear pore? Please try to rephrase this to clearer and simpler terms.

Line 405. There are a few places where strong terms like "unequivocally" are used, but you can rarely be that confident. For example, here, the unequivocal evidence is presumably contingent on TurboID-labelled NUP110 and NUP96 not having their sub-nuclear pore localisation perturbed by the TurboID tag?

Line 438. Use a term like "concentrated", not "focused".

Line 450. Same "single dot" problem as above. The images show ~20 dots.

Line 454. Is the signal a string or a bone? Use a single clear term.

Line 463. "due to phase separation" is not a valid reason here. One model of nucleolar formation is phase separation, but that's certainly not the mechanism maintaining the nucleolus structure in a fixed cell...

Line 493. This title is not accurate. You are taking known nuclear pore proteins (all based on localisation data from the TrypTag database, I believe?), and finding their likely position within the nuclear pore.

Line 604. You have not "assigned" proteins to a localisation, you have determined or discovered that they are on the cytoplasmic side.

Line 657. What is the evidence for this? How do you know that this is a "likely" cause?

Line 676. RabGAP in which species? What is the phylogeny of this family, and does that inform the evolution of the family?

Line 738. A balanced discussion of why previous immuno-EM data gave a symmetrical nuclear pore result is missing.

Line 739. The conclusion section has strong statements which need more balanced handling, particularly as it seems to be totally separated from discussion. For example this sentence describes the NUP76 cytoplasmic side result as "confidently" and "exclusively", when the actual result is that a tagged (ie. mutant) version of the protein is localised on that side.

Line 747. Significant should probably be reserved for statistical significance, again this is a sentence replete with strong statements without balanced discussion.

Line 758. This statement is not true. For example, a fenestrated nuclear envelope, without strong directed transport, may have evolved first - however no extant eukaryotes have been identified which retain this hypothetical ancestral state.

Line 760. This was a headline feature of the study, but you draw no specific conclusions.

Figure 1A. Indicate the protein being detected and the tagged terminus, ie. NUP76::HA and NUP110::TY (but correctly indicating N or C-terminal tagging, which is not clear).

Figure 1B. Accuracy of this plot is dependent on accurate expansion factor measurement, was that measured? How accurate is your expansion factor? These measurements for the tagged copies of the NUPs, and actual detected proteins (NUP76::HA, etc.) should be in the y axis labels.

Figure 1C-E. As for Figure 1A.

Figure 2A. Question marks in a figure are odd. Either you do or do not have evidence for these statements, please decide if that is the claim you are making. Also unclear if the small subtexts are intended to label that set of proteins or just one.

Figure 2B,C. The gene name is not "NUP75C", please use proper gene fusion names, ie. NUP75::TurboID and TurboID::NUP158, or otherwise label it clearly. Note that it is important and accurate to show that the prey proteins on the Y axis in A are the untagged wild-type versions.

Figure 1D. Worth noting in the corresponding text that you identified proteins identified as both cytoplasmic and nuclear side-specific, which is a good internal control against eg. biotin accessibility to the nuclear side.

Figure 2B,C. Gene fusion labelling is better, but could be a bit clearer.

Figure 4A. Why no clustering/dendrogram analysis, as in Figure 1A?

Figure 6B. Should show the data points.

Reviewer #2: 

The manuscript "A combination of expansion microscopy and proximity labelling reveals conserved and unique asymmetric functional hubs at the trypanosome nuclear pore" by Gabiatti et al. examine the organization of various nucleoporins within the nuclear pore complex of trypanosomes. Using expansion microscopy and mass spectrometry, the authors redefine the symmetry of previously identified nucleoporins, such as the Nup76 complex, within the structure of the trypanosome nuclear pore complex - the show that a fraction of nucleoporins are asymmetric localized within NPCs as in other organisms and congruent with their assumed role in directed nuclear transport events through the NPC. They also explore the nucleocytoplasmic transport machinery associated with NPCs, focusing on the localization of different transport factors involved specifically in mRNA transport. The proximity based mass spectrometric analysis uncovered several previously unidentified proteins that may play significant roles in nucleocytoplasmic transport and warrant further investigation. 

Overall, I find this study very interesting and technically of very high standard. It provides a huge quantity of relevant data and will significantly advances our knowledge about NPCs in the trypanosome, important also as an organisms outside the ophistokonts for insightful evolutionary comparisons. Nevertheless, I see some minor points regarding the robustness of its conclusions and the extent to which the findings can be confidently interpreted based on the experimental methods employed, which I hope the authors can easily address.

Specific points:

1.) Line 325-326 (also materials and methods) "Upon dual labelling with anti-Ty1 and anti-HA, we carried out expansion and imaging": Why do the authors do the labeling before the expansion? It was described in literature by others that the antibody labeling is done following the expansion. Does this make any difference?

2.) Line 327-329 "The signals from the NUP76-complex proteins were in all cases clearly separated from the NUP110 signal towards the cytoplasmic site of the pore (Figure 1A and Figure S2 in supplementary material).": The data indeed show that Nup76, Nup140 and Nup149 are separated from Nup110. Why did the author chose Nup110 instead of a more central localized nucleoporin such as Nup64 (used in figure 1E)? As from figure 1B, it is difficult to get a feeling of how close/far are Nup76, Nup140 and Nup149 from each other which might be due to relating those measurements to the distant Nup110 instead of a more central Nup.

3.) Line 329-331: "Notably, we observed for every dot signal originating from the NUP76 complex a corresponding NUP110 dot, indicating that trypanosomes, unlike yeast (Galy et al, 2004), do not have basket-less pores.": Why this is not the case for Nup140 and Nup149? Does this represents variabilities in the NPC or is it a result of inefficient labeling?

4.) Figure 2. A proximity map of the trypanosomatid nuclear pore: It is inetersting that Nup76 is seemingly absent at the nuclear ring side. Did the authors try to use other members of the Nup76 complex such as Nup140 and Nup149 as a bait? There seems to be a decent signal for Nup149 in proximity with C-terminal turboID-labeled Nup110.

5.) Line 457-459: "For Ran, Mex67 and RanBPL, we observed an additional signal at the nucleolus, which is defined by the reduction in DAPI stain (Figure S6A), and a minor signal in the nucleoplasm.": Why is it that this bone like signal happens at some NPCs but not others. The RAN images in figure 3C do not align well to the results for RAN in figure 3A where there is no signal detected at all at the nuclear ring side.

6.) Line 494-496: "To predict the localisation of the remaining 38 nuclear pore-localised proteins more accurately, we extended the NUP76/96/110 map by the proximity labelling data of Mex67 and Ran and of the outer ring protein NUP158 (Moreira et al, 2023).": Given that Nup110, Nup96 and Nup76 proximity labeling have failed to fully predict RAN and Mex67 binding. Can this analysis here be trusted to reveal the true localization of those 38 proteins? Given that many of them are shuttling nuclear transporter receptors/factors that will probably behave in a similar way to Mex67 and RAN. This might also explain why some unknown proteins such as Tb927.11.1000, Tb927.10.12200, Tb927 and Tb927.10.8160. Can expansion microscopy be used here to reveal the localization some of those unknown proteins?

7.) Line 655-660: "Of the (putative) NUP76 associated proteins, only the pore localisation of NUP140 was clearly abrogated upon NUP76 depletion, while NUP149 and Gle2 still localised to the pore. Note that a slightly diminished pore localisation was observed for all four proteins, likely caused by the disrupted mRNA export and general loss in fitness rather than a specific impact on nuclear pore architecture. Thus, NUP140 localisation to the pore is fully dependent on NUP76, while NUP149 and Gle2 appear to be anchored independent of NUP76.": This is an very interesting finding as in mammals a Nup88 knockdown abolishes Nup214 NPC localization but would not have the same extensive effect on Nup62, which also depends on other nucleoporins for NPC localization. 

8.) Figure S2B: the labeling pattern (outside versus inside) seems to be reverted in the upper as compared to the two other panels. Can the authors comment on this?

Reviewer #3: 

The authors use a combination of state of the art techniques to better characterize the nuclear pore in Trypanosoma brucei. The apply different expansion microscopy techniques to demonstrate that different than previously thought the nuclear pore of T. brucei actually shows some asymmetries that help to understand the mRNA transport directionality in this system. Furthermore they apply proximity labelling together with mass spectrometry to identify novel components of the pore and verify subcomplex composition of this very important and eukaryote defining structure of the nuclear pore.

From my perspective the major contributions that come from this manuscript are:

1. Establishing the asymmetry of the pore also in Trypanosomes

2. Identification of novel components and verification of subcomplex compositions

The weaker part of the manuscript concerns the functional role of NUP67. The experiments do show that the protein is involved in mRNA export but the statement that it is the functional orthologue of NUP82/88 is in my opinion an overstatement and also not necessary. Using the degron system is nice but I am not sure why RNAi would not also have worked here? 

The manuscript is overall ok written. I am not a big fan of combining results and discussion this is especially problematic if the manuscript contains much bioinformatics data where there is notoriously a fine line between results and discussion. 

I strongly suggest to restructure the manuscript such that the results and discussion are separated.

Also many of the figure legends contain results statements. They should be removed.

This aside I congratulate the authors to a great piece of work that will spike interest in a larger audience and should be published.

Minor comments:

The authors write: Instead, nuclear export is likely fuelled by the GTP/GDP gradient created by the Ran GTPase. The authors speak of the Ran GTPase? is there only one in the T. brucei genome? please be specific here.

This is a rather convoluted sentence: The resulting availability of reference proteins for basket, inner ring and cytoplasmic site allowed mapping of all 75 trypanosome proteins with known nuclear pore localisation to a sub-region of the pore based on mass spectrometry data from proximity labelling. What exactly do the authors mean here?

Again the following sentence is not really clear: This approach defined many further asymmetrically localised nuclear pore components. I assume they mean to define additional proteins that are asymmetrically localized at the nuclear pore?

Confirmation of absence is quite a challenging task. I would tone down the statement that you confirmed the absence of Dbp5.

The authors should inform that they do not know if the proteins that they have tagged are actually functional and thus might have changed localization it is important to mention this. 

The authors state that they express from the endogenous locus and thus do not change expression levels. Key for this is the presence 3 prime UTR of the tagged gene, was this maintained? If not please clarify.

The following statement is too strong and not supported by the data: This phenotype is similar to the one observed upon Nup82 depletion in yeast (Hurwitz & Blobel, 1995; Grandi et al, 1995), strongly suggesting that NUP76 is the functional orthologue to yeast Nup82 with a crucial role in mRNA export. I am sure the depletion of many components of the nuclear pore will lead to a lack of mRNA export. I think the authors need include further support for their claim.

Figure 6: 

The authors make a great effort to characterize the organization of the pore by expansion microscopy using the relative positioning of individual components. But in the degron experiment they claim the pore assembly is not affected by simply showing one components, which could be different in its localization upon the depletion of NUP76 affecting the pore function indirectly. In order to make a solid claim here the authors could one or two components they analysed in detail above and show their relative localization does not change. This would be much stronger evidence.

The authors state: Notably, we observed for every dot signal originating from the NUP76 complex a corresponding NUP110 dot, indicating that trypanosomes, unlike yeast (Galy et al, 2004), do not have basket-less pores. 

The authors could discuss this point? does this mean there are no assembly intermediates?

corrections:

Altogether, the combination of proximity labelling with expansion microscopy revealed an asymmetric architecture of the trypanosome nuclear pore supporting inherent roles fort directed transport. "for" instead of fort.

...but with a fundamental different mRNA remodelling platform at... fundamentally

---

## [Editor Report · Decision Letter 2]

13 Jan 2025

Dear Dr Kramer,

Thank you for your patience while we considered your revised manuscript "A combination of expansion microscopy and proximity labelling reveals conserved and unique asymmetric functional hubs at the trypanosome nuclear pore" for publication as a Research Article at PLOS Biology. This revised version of your manuscript has been evaluated by the PLOS Biology editors, and the Academic Editor.

Based on our Academic Editor's assessment of your revision, we are likely to accept this manuscript for publication, provided you satisfactorily address the following editorial requests. Please also make sure to address the following data and other policy-related requests.

a) We routinely suggest changes to titles to ensure maximum accessibility for a broad, non-specialist readership, and to ensure they reflect the contents of the paper. Although we understand that the methods allowed the new discovers, the study is not a Methods and Resources paper and we think they are not the key contributions of the study. We would like to suggest the following. Please ensure you change both the manuscript file and the online submission system, as they need to match for final acceptance:

"Detailed characterization of the trypanosome nuclear pore architecture reveals conserved asymmetrical functional hubs that drive mRNA export"

Please supply the numerical values either in the a supplementary file or as a permanent DOI’d deposition for the following figures:

Figure Fig 2A, 3A, 4A, 6B, S4, S5A, S7, S12B, S13B, S14B

c) Please cite the location of the data clearly in all relevant main and supplementary Figure legends, e.g. “The data underlying this Figure can be found in S1 Data” or “The data underlying this Figure can be found in https://doi.org/10.5281/zenodo.XXXXX”

d) We require the original, uncropped and minimally adjusted images supporting all blot and gel results reported in the Figures 6A, S5B, S11

We will require these files before a manuscript can be accepted so please prepare and upload them now. Please carefully read our guidelines for how to prepare and upload this data: https://journals.plos.org/plosbiology/s/figures#loc-blot-and-gel-reporting-requirements

e) Please note that by journal policies, all data should be provided either as Supplementary Material or on a repository such as Zenodo. Anything that is not in the Supplementary Material should be deposited in a stable, community-accepted repositories, or somewhere like zenodo, and the accession numbers provided.

Thank you for providing additional microscopy pictures for some of the figures. If you would like to provide more pictures, we would like to suggest that you could also upload all Microscopy images present in Figures 1A-D, 3BC, 6CDE, S2ABC, S3, S6AB, S8B, S12A, S13A, S14A, S14, to the "image data resource", https://idr.openmicroscopy.org/about/index.html - information on how this can be done can be found here https://idr.openmicroscopy.org/about/submission.html.

f) Per journal policy, if you have generated any custom code during the course of this investigation, please make it available without restrictions upon publication. Please ensure that the code is sufficiently well documented and reusable, and that your Data Statement in the Editorial Manager submission system accurately describes where your code can be found.

We expect to receive your revised manuscript within two weeks. 

*Published Peer Review History*

*Press*

Sincerely,

Melissa

Melissa Vazquez Hernandez, Ph.D.

Associate Editor

PLOS Biology

---

## [Editor Report · Decision Letter 3]

17 Jan 2025

Dear Susanne,

Thank you for the submission of your revised Research Article "Detailed characterization of the trypanosome nuclear pore architecture reveals conserved asymmetrical functional hubs that drive mRNA export" for publication in PLOS Biology. On behalf of my colleagues and the Academic Editor, Ándré Schneider, I am pleased to say that we can in principle accept your manuscript for publication, provided you address any remaining formatting and reporting issues. These will be detailed in an email you should receive within 2-3 business days from our colleagues in the journal operations team; no action is required from you until then. Please note that we will not be able to formally accept your manuscript and schedule it for publication until you have completed any requested changes.

PRESS

Sincerely, 

Melissa

Melissa Vazquez Hernandez, Ph.D., Ph.D.

Associate Editor

PLOS Biology
